# Basonuclin-2 regulates extracellular matrix production and degradation

Ayla Orang[1], B Kate Dredge[1] , Chi Yau Liu[1], Julie M Bracken[1], Chun-Hsien Chen[1], Laura Sourdin[1] , Holly J Whitfield[4,5] , Rachael Lumb[1], Sarah T Boyle[1], Melissa J Davis[3,4,5,6,7], Michael S Samuel[1,8] , Philip A Gregory[1,2], Yeesim Khew-Goodall[1,2], Gregory J Goodall[1,2] , Katherine A Pillman[1,2,*] , Cameron P Bracken[1,2,*]

Epithelial–mesenchymal transition is essential for tissue patterning and organization. It involves both regulation of cell motility and alterations in the composition and organization of the ECM—a complex environment of proteoglycans and fibrous proteins essential for tissue homeostasis, signaling in response to chemical and biomechanical stimuli, and is often dysregulated under conditions such as cancer, fibrosis, and chronic wounds. Here, we demonstrate that basonuclin-2 (BNC2), a mesenchymal-expressed gene, that is, strongly associated with cancer and developmental defects across genome-wide association studies, is a novel regulator of ECM composition and degradation. We find that at endogenous levels, BNC2 controls the expression of specific collagens, matrix metalloproteases, and other matrisomal components in breast cancer cells, and in fibroblasts that are primarily responsible for the production and processing of the ECM within the tumour microenvironment. In so doing, BNC2 modulates the motile and invasive properties of cancers, which likely explains the association of high BNC2 expression with increasing cancer grade and poor patient prognosis.

## Introduction

The interchangeable transformation between cells displaying epithelial or mesenchymal characteristics (epithelial–mesenchymal transition [EMT]) facilitates both the migration of cells in a mesenchymal state and the subsequent re-establishment of an epithelium. This is required at multiple stages during development, including gastrulation, the migration of neural crest cells, and in the formation of structures such as heart valves and kidney tubules.

Along with the reversion back to epithelial form (MET), EMT then continues to play important roles in the adult, one example being tissue repair where migration across a wound and subsequent re-epithelialization is required for healing. However, this same process that facilitates motility can also be commandeered by cancer, with EMT/MET facilitating the local invasion of cancer cells and each of the subsequent steps of intravasation, extravasation, and secondary tumour growth that are associated with metastatic progression (1, 2).

One of the most central aspects of EMT is the critical role it plays in guiding the formation and organization of the ECM; a complex network of proteins (known as the "matrisome") that provides an anchorage point for cells that regulates the activation of signaling pathways driven by biomechanical and chemical signals and whose composition is altered within the tumour microenvironment (3, 4). Coupled with elevated stemness and chemoresistance, that is, bestowed by mesenchymal gene expression, it is not surprising that patients whose tumours have higher EMT signatures generally have worse prognosis (5). Because of the importance of EMT and the cancer-associated properties of mesenchymal cells, gaining a better understanding of EMT, the ECM and novel regulators of these processes are of significant interest. In this study, we interrogate the function of one such potential new regulator, basonuclin-2 (BNC2), initially selected on the basis of its mesenchymal-specific expression and its predicted direct repression by miR-200; a potent enforcer of the epithelial phenotype and a critical regulator of the E/M status of cells (6, 7).

With the exception of one in silico study in which BNC2 was proposed to drive a mesenchymal state solely on the basis of its co-expression with other genes (8), nothing has been reported specifically linking BNC2 to EMT. However, a number of reports do implicate this gene in various important processes, including cancer. Single-nucleotide polymorphisms (SNPs) in the physical

[1]Centre for Cancer Biology, An Alliance of SA Pathology and University of South Australia, Adelaide, Australia   [2]Department of Medicine and School of Biological Sciences, University of Adelaide, Adelaide, Australia   [3]South Australian ImmunogGENomics Cancer Institute, Faculty of Health and Medical Sciences, University of Adelaide, Adelaide, Australia   [4]Division of Bioinformatics, Walter and Eliza Hall Institute of Medical Research, Parkville, Australia   [5]Department of Medical Biology, Faculty of Medicine, Dentistry and Health Sciences, University of Melbourne, Parkville, Australia   [6]Department of Clinical Pathology, Faculty of Medicine, Dentistry and Health Sciences, University of Melbourne, Parkville, Australia   [7]Fraser Institute, University of Queensland, Wooloongabba, Australia   [8]Adelaide Medical School, Faculty of Health and Medical Sciences, University of Adelaide, Adelaide, Australia

Correspondence: cameron.bracken@unisa.edu.au; katherine.pillman@unisa.edu.au
*Katherine A Pillman and Cameron P Bracken contributed equally to this work

proximity of the BNC2 gene were identified by genome-wide association study (GWAS) as having the highest association with ovarian cancer risk ([9]). Subsequent mapping studies reaffirmed this observation, repeatedly identifying SNPs ([10], [11], [12], [13]), not within the protein-coding region of BNC2, but at sites that are either intronic or within a 44 kb region upstream of the BNC2 transcription start site. Coupled with observations that BNC2 expression is lower in ovarian cancer cell lines compared with normal tissue ([14]) and that the deletion of SNP-containing regions decreases expression ([11]), BNC2 was thereby implicated as an important cancer-associated gene. Similar studies also note the association of BNC2-related SNPs across a range of cancer types (esophageal ([15]), prostate ([16]), bladder ([17], [18], [19]), liver ([20]), skin ([21], [22], [23]), uterus ([24], [25]), stomach ([26]), and lung ([27])).

Developmental functions are also indicated through deletion or mutation of the *BNC2* locus. In mice, BNC2 knockout is typically embryonic lethal, with survivors dying shortly after birth with craniofacial abnormalities, consistent with BNC2 expression in mesenchymal cells of the palate and mesenchymal sheaths surrounding cartilage and bone ([28]). BNC2 expression is induced during cartilage differentiation in the developing arm ([29]) and surviving BNC2−/− mice display dwarfism, leading to the suggestion that BNC2 may be required for the survival or development of chondrocytes ([30]). This in turn may explain extensive GWAS-linking *BNC2* with adolescent idiopathic scoliosis ([31], [32], [33], [34], [35], [36]). Body curvature is reported in BNC2 knockout zebrafish that also display developmental defects of the urinary tract ([37]), another phenotype linked to BNC2 by human GWAS ([38], [39]). BNC2−/− mice also possess shorter hair follicles, leading to the characterization of BNC2 expression in mesenchymal stem-like hair follicle progenitor cells ([30]). Lastly of note, a mutagenesis approach in zebrafish linked BNC2 to a defective striping pattern, where it was then discovered that BNC2 is required for the survival and development of chromatophore cells that are responsible for pigmentation ([40], [41]). Again, this is reflected in extensive human GWAS that have implicated BNC2 with skin tone and freckling ([42], [43], [44], [45], [46], [47], [48], [49], [50]). A diverse array of phenotypic effects are thus linked to the perturbation of BNC2; however, these may still point to common mechanisms in cell survival, migration, differentiation, or some aspect of EMT that manifests differently depending upon the nature of BNC2 dysregulation and the developmental or cancer context being examined.

In this study, we sought to explore the role of BNC2 in mesenchymal breast cancer cells in which it is naturally highly expressed. We find BNC2 is not simply another general promoter of the mesenchymal phenotype, but instead regulates the composition and digestion of the ECM, which is consistent with the migratory, invasive phenotype that endogenous BNC2 promotes in cancer cells. This is also consistent with a very recent report in which BNC2 was associated with myofibroblast activation in fibrotic livers ([51]).

Here, we report that BNC2 is most highly expressed in tissue fibroblasts, the cell type, that is, primarily responsible for the constitution of the ECM, where BNC2 regulates the level of secreted collagens and metalloproteases and the capacity of cells to digest a gelatin substrate. The ECM as a whole is neither oncogenic nor tumour suppressive, with extensive reports of both roles in cancer depending upon context. Likewise, we note an intriguing paradox whereby reduced BNC2 expression is generally associated with tumours; however, very high BNC2 expression is associated with poor patient prognosis and higher tumour grade. We hypothesise that the extensive regulation of ECM composition and digestion, that is, controlled by BNC2 in both cancer cells and fibroblasts is central to its frequently reported role in cancer and development, but which until now has remained largely mechanistically uncharacterised.

# Results

## BNC2 is a mesenchymally expressed, EMT-responsive gene, that is, targeted by miR-200

Whether a cancer has predominantly epithelial or mesenchymal features can have a profound effect on patient outcome, with mesenchymal characteristics promoting both the invasiveness of cells and stem-like, chemoresistant properties ([1], [2]). In an effort to characterize such tumours, individual breast cancer patient biopsies were scored against an E/M gene expression signature ([52]), which indicated that tumours with high BNC2 expression are more mesenchymal in their nature (Fig 1A–D). This is the case regardless of breast cancer subtype (Fig 1E) and is true, not only across primary tumours but also across breast cancer cell lines (Fig 1F). Consistent with a gene whose expression is predominantly mesenchymal, BNC2 expression is progressively increased in epithelial cells subjected to treatment with EMT-inducing TGF-β (Fig 1G–I). Unfortunately, the measurement of BNC2 expression is limited to mRNA as, despite published and commercially available antibodies, we have not been able to demonstrate that any are reliably capable of detecting BNC2 protein, both at endogenous levels and when tagged versions are expressed (Fig S1). Consistent with a mesenchymally expressed gene, and with that of a potential EMT-driver, the 3′UTR of BNC2 also possesses predicted target sites for both targeting classes of the miR-200 family (miR-200b/200c/-429 and miR-200a/-141) (Fig 1J), especially miR-200b/-200c/-429 for which there are three putative sites (two very confidently predicted), placing it within the top 2.5% of all miR-200c predicted targets (TargetScan algorithm, ([53])). Accordingly, miR-200c decreases the expression of endogenous BNC2 mRNA and the activity of a BNC2-3′UTR luciferase reporter (Fig 1K and L).

## BNC2 is not a general promoter of EMT

BNC2 is widely believed to function as a transcription factor (TF) based on its reported nuclear localization and the presence of four C2H2-type Zn-finger motifs, reminiscent of other DNA binding proteins. Coupled with its mesenchymal expression, induction in response to TGF-β and targeting by miR-200 (Fig 1), we initially hypothesized that BNC2 is likely to directly drive a mesenchymal transcriptional program, reminiscent of well-characterized EMT-promoting TFs such as PRRX1 and members of the ZEB, SNAIL, and TWIST families ([54]). Accordingly, BNC2 expression is positively correlated with these genes and with other markers of mesenchymal cells across cancer cell line encyclopedia (CCLE) and breast

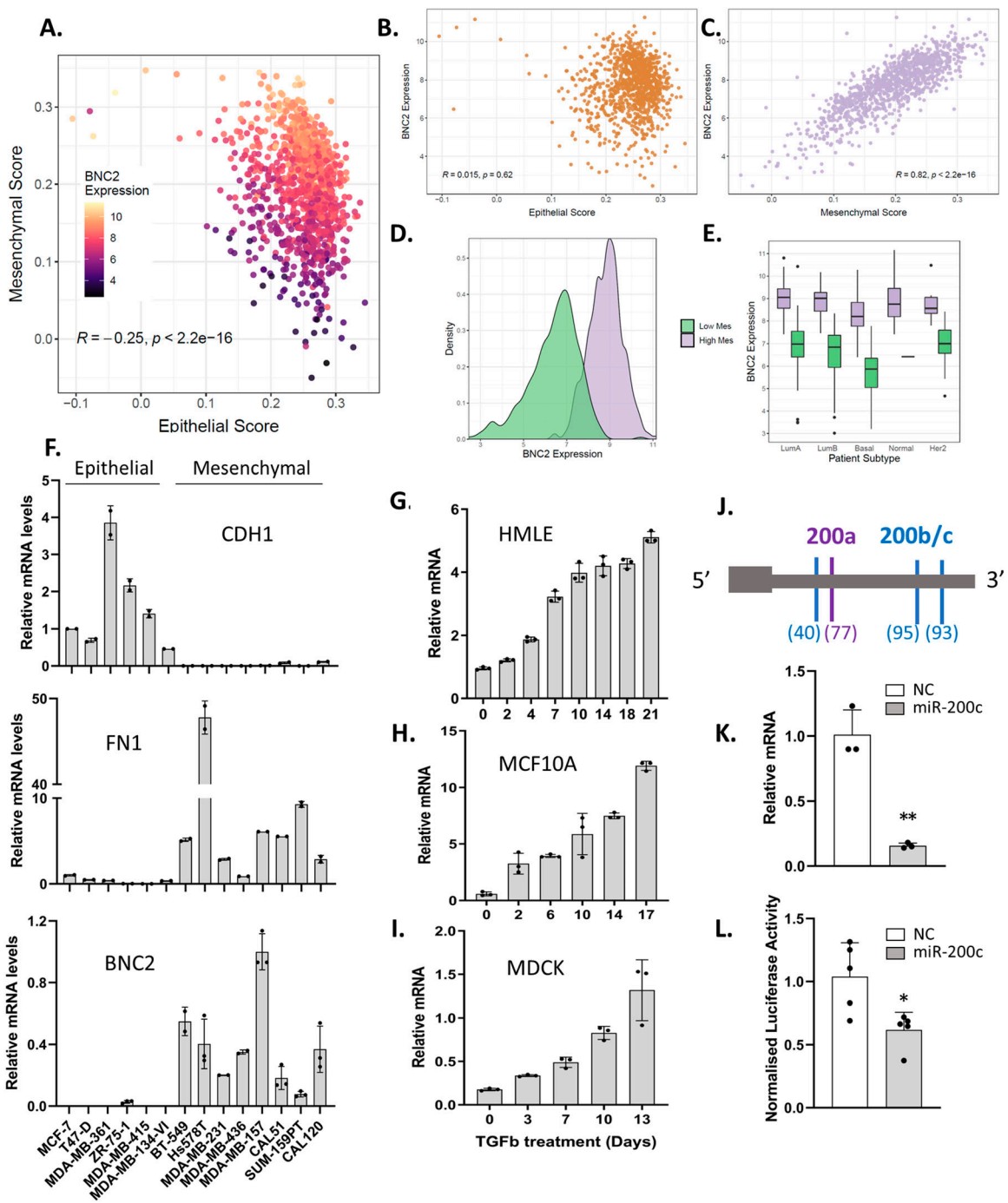

**Figure 1. Basonuclin-2 (BNC2) is a mesenchymally expressed, epithelial–mesenchymal transition responsive gene, that is, targeted by miR-200.**
**(A)** An epithelial or mesenchymal score is attributed to individual tumours (breast cancer, TCGA) based upon previously established gene signatures (52). **(B, C)** BNC2 expression is labelled for all tumours on an epithelial versus mesenchymal axis, or when epithelial (B) or mesenchymal (C) genes are considered separately. **(D)** BNC2 expression across breast cancer samples (TCGA) stratified by high versus low mesenchymal score. **(E)** BNC2 expression in TCGA samples across breast cancer subtypes, stratified by mesenchymal score. **(F)** BNC2 expression as measured by qRT-PCR across a panel of human breast cancer cell lines. CDH1 (E)-cadherin is an epithelial marker, FN1 (fibronectin) is a mesenchymal marker. **(G, H, I)** Relative BNC2 expression upon continuous TGF-$\beta$ treatment of multiple epithelial-like cell lines (G) HMLE; (H) MCF10A; (I) MDCK. **(J)** Predicted miR-200–responsive target sites within the BNC2 3′UTR. Scores (in brackets) for miR-200a or miR-200b binding are taken from TargetScan 7.1, in which the most confidently predicted sites are indicated by scores closer to 100. **(K)** BNC2 expression as measured by qRT-PCR in response to miR-200c in mesHMLE cells. **(L)** Normalized activity of a BNC2-3′UTR Renilla/luciferase reporter gene in response to miR-200c. All data are representative of triplicate experiments. * indicates $P < 0.05$, **$P < 0.01$, ***$P < 0.001$ (unpaired $t$ test) relative to control transfection.

cancer patients (TCGA) (Fig S2). Conversely, BNC2 expression negatively correlates with epithelial markers such as CDH1, ESRP1, and ESRP2.

To test our hypothesis that BNC2 drives a mesenchymal phenotype, BNC2 was knocked down in three breast cancer cell lines (BT549, HS578T, and mesHMLE) in which it is relatively highly expressed (Fig 2A–C) or was induced in MCF7 cells in which it is not normally present (Fig 2D and E). Expression of genetic markers of mesenchymal or epithelial status was then assessed by qRT-PCR. Other than a consistent reduction in ZEB2 upon BNC2 knockdown, common markers of EMT were either unaffected, or only modestly altered, but in a cell-specific manner that was not consistent with the maintenance or enforcement of a mesenchymal phenotype as we had hypothesized. Similarly, if BNC2 knockdown were to drive MET, one would anticipate consistent trends in the expression of the same genes used to score the epithelial or mesenchymal characteristic of tumours in Figs 1A and 2F. No such trend was observed. Furthermore, BNC2 perturbation had no obvious effect on cell morphology (Fig S3) or on the distribution of actin or other EMT markers as examined by immunofluorescence (Fig S4). Although we cannot discount a role for BNC2 as a promoter of some aspect of a hybrid E/M state, these observations collectively argue against BNC2 directly driving a mesenchymal phenotype, despite itself being highly mesenchymal-specific in its pattern of expression.

## BNC2 is a core component of an EMT and ECM gene expression module

Although the strong correlation between BNC2 gene expression and known markers of EMT clearly demonstrates mesenchymal associations (Fig S2), EMT represents a broad array of phenotypic features, many of which are not necessarily captured through the profiling of specific endpoint genes or EMT drivers. As such, we performed a weighted gene co-expression network analysis (WGCNA) which defines modules of genes that are differentially co-expressed across samples, in this case ~1,200 breast cancer primary tumours from TCGA. BNC2 was determined to be a strong "hub" gene for a large co-expression module (arbitrarily designated "red," Fig 3A), being the fifth (of 497 genes) most highly correlated with other genes in the module. The function of genes in the BNC2-containing "red" module is enriched in EMT- and ECM-associated processes as indicated by gene ontology (GO) analysis (Figs 3B and S5).

## Endogenous BNC2 regulates ECM composition and degradation

WGCNA indicates endogenous BNC2 expression is correlated with that of EMT and ECM-associated genes. To identify genes whose expression is dependent upon BNC2, endogenous BNC2 was knocked down by siRNA in BT549 cells, the breast cancer cell line that displayed the highest levels of BNC2 expression within the CCLE, Depmap, (55). BNC2 knockdown strongly influenced the expression of genes associated with EMT and several related functions that were identified in the patient WGCNA, including the same ECM, collagen, and collagen catabolism processes (Figs 4A–E and S6). This was particularly the case with a cohort of fibrillar-type collagens (type I, III, V, XII), up-regulated by BNC2 knockdown, and specific matrix metalloproteases (MMPs) which were substantially

down-regulated upon BNC2 depletion (Fig 4F). Similar expression patterns of collagens and MMPs in response to BNC2 depletion was also seen in HS578T cells, another mesenchymal-like breast cancer cell line that expresses abundant BNC2, and also in mesHMLE cells, an immortalized epithelial breast cell line driven toward a mesenchymal phenotype by 3 wk of TGF-β exposure (Fig 4G). Accordingly, production of both collagen type I and type V proteins are strongly elevated after BNC2 knockdown in both BT549 and HS578T cells (Fig 5A–C). Collectively, the identification of the genes with which BNC2 is co-expressed in cancer patients (Fig 3, "red" module), and the matrisomal genes whose expression is affected by BNC2 perturbation (Fig 4), indicate that BNC2 may serve as a novel regulator of ECM composition and processing.

## Endogenous BNC2 controls collagen production and matrix degradation in fibroblasts

Although BNC2 strongly influences collagen production in multiple breast cancer cells, immunofluorescence analysis of collagens in cells is not a clear representation of the collagen composition of the ECM (Fig 5). Furthermore, it is not breast cancer cells but cancer-associated fibroblasts that are primarily responsible for the production and remodeling of ECM proteins in tumours (56, 57). With this in mind, we sought to ascertain whether BNC2 is also a regulator of the ECM in fibroblasts, especially as it is in fibroblasts that BNC2 is most abundantly expressed across tissues (Fig 6A). Similarly, to breast cancer cells, BNC2 knockdown in mammary fibroblasts strongly down-regulates the expression of select MMPs and up-regulates the expression of specific collagens at both the RNA and protein levels (Fig 6B and C). Immunofluorescence analysis of fibroblast cell–derived matrix (CDM) also confirms that BNC2 knockdown in fibroblasts increases the level of secreted collagen I (Fig 6D). Furthermore, MDA-MB-231 mammary tumour cells seeded into the CDM derived from fibroblasts in which BNC2 had been knocked down exhibited a trend towards reduced motility relative to those seeded into CDM derived from fibroblasts that had been transfected with control non-targeting siRNA (Fig 6E). Consistent with Mmp1 repression, BNC2 knockdown decreased the capacity of fibroblast cells to degrade a fluorescent gelatin matrix (Fig 6F–H). Taken together, these data suggest that BNC2 regulates the production and degradation of the ECM, not only within cancer cells themselves but also in cancer-associated fibroblasts that are fundamental to establishing the tumour microenvironment.

## BNC2 promotes cancer cell motility and invasion

The composition of the matrisome and the manner in which it is modified and degraded is of particular relevance to the motility and invasive properties of cells and thus, to their metastatic potential. Consistent with an enhanced capacity of mesenchymal cells to migrate, BNC2 knockdown across a number of cell lines (BT549, HS578T, and mesHMLE) decreased motility, invasion, and wound healing, and did so consistently with two independent siRNAs and an inducible short hairpin construct in stable independent clones (Figs 7A–H, S7, and S8). Conversely, Dox-inducible expression of BNC2 in epithelial cells (MCF7 and T47D cells that otherwise express very little endogenous BNC2) increased wound closure in

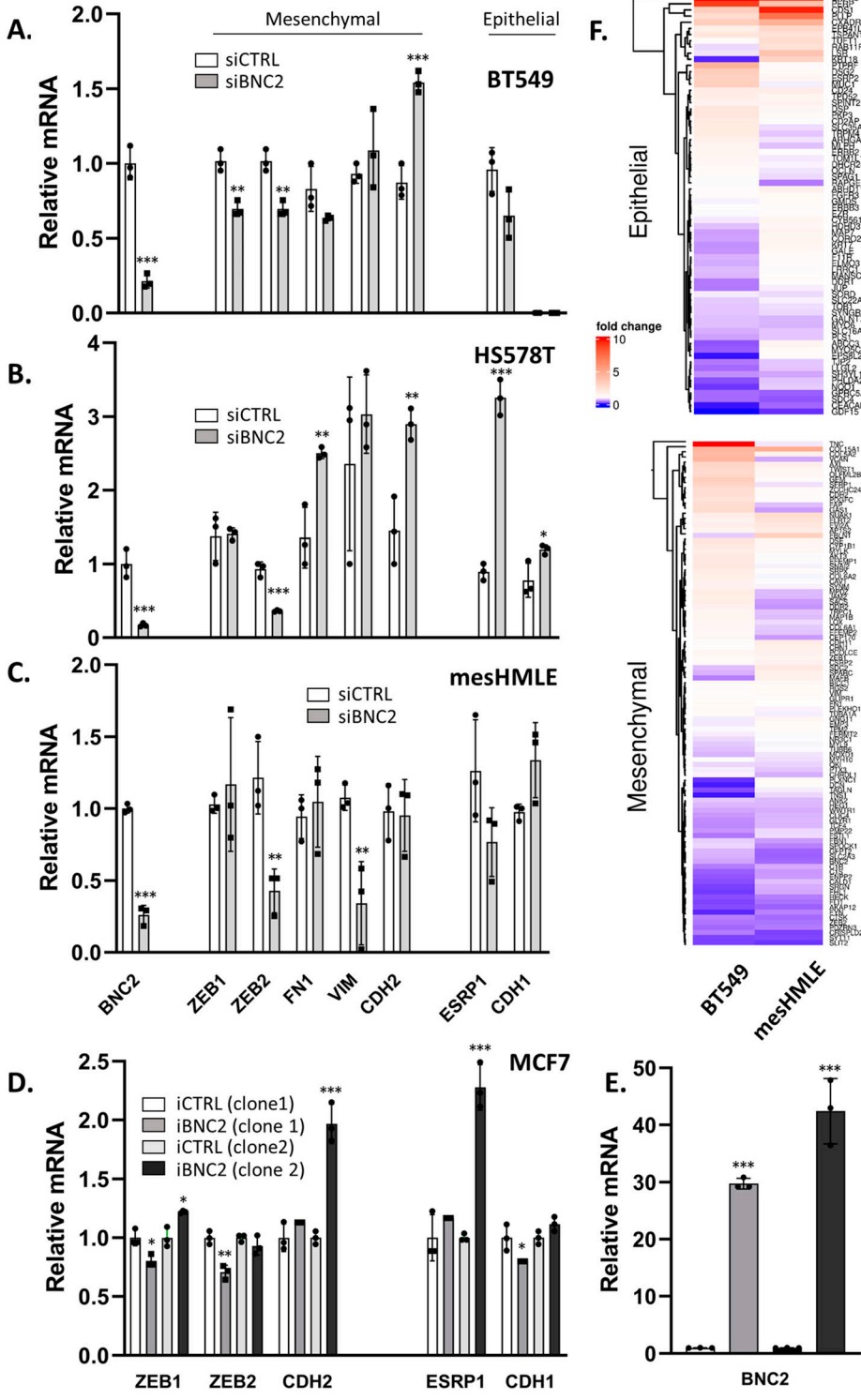

**Figure 2. Basonuclin-2 (BNC2) is not a core regulator of epithelial–mesenchymal transition.**

**(A, B, C)** Gene expression was measured by qRT-PCR after two rounds of transfection of BNC2 siRNAs across 6 d in (A) BT549, (B) HS578T, and (C) mesHMLE cells. **(D)** Gene expression was measured after Dox-mediated induction of a stably introduced BNC2 full-coding sequence in two MCF7 clonal cell lines. Gene expression of mesenchymal and epithelial markers are indicated. **(E)** BNC2 RNA induction upon Dox treatment. All data are representative of multiple experiments. **(A, B, C, D)** * indicates $P < 0.05$, **$P < 0.01$, ***$P < 0.001$ (unpaired $t$ test) relative to either control transfection (A, B, C) or no Dox (D). **(F)** Fold change as assessed by RNA-Seq in the expression of epithelial or mesenchymal marker genes after BNC2 knockdown in BT549 and mesHMLE cells. From the 146 epithelial–mesenchymal transition signature genes, all genes that are expressed at >1 cpm in both cell lines are shown.

independent clonal cell lines after 3 d (Figs 7I–L, S7, and S8). To check that the effects on motility/invasion are not biased by cell number (that may alter on account of changes in proliferation, apoptosis etc.), crystal violet assays were performed (Fig S9). SiRNA knockdown of BNC2 did result in a small decrease in cell number,

but far less than the effect size of BNC2 knockdown with regard to migration, invasion, and wound closure. ShRNA and Dox-inducible BNC2 had no effect on cell number, although each had significant effects on wound closure. These data, coupled with functional data from fibroblasts (Fig 6), suggest that cancer cell motility is regulated

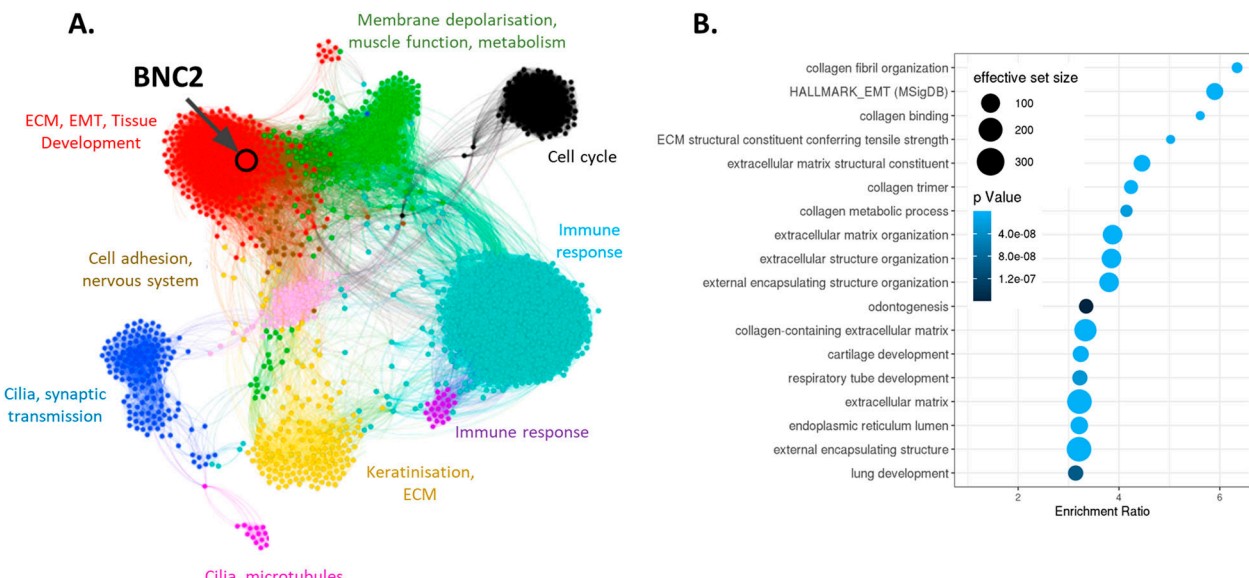

**Figure 3. Basonuclin-2 is a central component of an epithelial–mesenchymal transition and ECM gene expression module.**
**(A)** Graph visualization of a network derived from weighted gene co-expression network analysis performed using the 5,000 most differentially expressed genes among TCGA breast cancer patients. The co-expression modules identified (shown as colours) are labelled with general functions based upon GO analysis. Basonuclin-2 was the fifth most highly connected gene within the "red" cluster. **(B)** GO analysis results for the 170 most centrally associated genes (correlation with module eigengene of >0.7) within the "red" co-expression module.

by BNC2 via two independent but converging mechanisms; directly by altering the nature of the cancer cells themselves (for those in which BNC2 is naturally expressed), and indirectly through cancer-associated fibroblasts that produce and remodel the ECM.

### BNC2 expression correlates with more aggressive cancer

The results presented thus far are consistent with those of a gene whose expression promotes cancer progression; it is highly mesenchymal and its perturbation indicates a positive role in driving cell invasion. Seemingly in contradiction, however, BNC2 has been proposed to act as a likely tumour suppressor as it exhibits lower expression in ovarian cancer cell lines than in non-cancerous ovarian cells (14). We also find BNC2 is down-regulated in 9 of 28 cancer types, including breast cancer (Fig 8A), and trends toward down-regulation in a further seven, whereas up-regulation is observed in six cancer types, only three of which being statistically significant (Fig S10A). Despite generally lower levels within the bulk tumour, BNC2 expression increases with cancer stage (Figs 8B and S10B) and elevated BNC2 expression correlates with lower survival (Figs 8C, S10C, and S11). Prognosis becomes notably worse in patients whose level of BNC2 expression is highest (Figs 8C and S10C; compare red and blue lines in upper/lower 10th percentiles, quartiles, and halves (90/10; 75/25, and 50/50).

Critical to cancer outcome is not only the nature of the cancer cells themselves but also the tumour microenvironment in which they reside. As the microenvironment is largely shaped by the actions of fibroblasts (59), and endogenous BNC2 is most broadly and highly expressed in fibroblasts (Fig 6), we sought to question in which population of tumour-associated cells that BNC2 is expressed most highly. BT549 and HS578T breast cancer cells were

selected for previous experiments, specifically because of their high expression of BNC2. However, wider examination of the CCLE indicates these are outliers as in most of the breast cancer cell lines, BNC2 is either lowly expressed or absent (Fig 8D). This is in contrast to the uniformly high level of BNC2, that is, expressed across fibroblasts (Fig 8D). As we do not possess an antibody that unequivocally detects endogenous BNC2 (Fig S1), we examined single-cell sequencing derived from 26 breast cancer patients (>130 K single cells) to determine cell-specific BNC2 expression (58). We find that BNC2 is almost exclusively expressed in cancer-associated fibroblasts, especially those of the s5 myofibroblast-like subtype (Fig 8E) and was absent in the epithelial tumour cells themselves (at least in this cohort of patients). BNC2 expression closely paralleled that of the fibroblast marker FAP. Notably, the exact same trend of poor prognosis was observed for the patients who expressed the highest levels of FAP (Figs 8F and S10D) as was observed for patients expressing the highest levels of BNC2. Collectively, these data are consistent with cancer-associated fibroblasts being the primary source of BNC2-expressing cells in tumours. The poor prognosis of patients expressing the highest levels of BNC2, and the elevated expression of BNC2 at more advanced cancer stages, may be more indicative of infiltrating fibroblasts than it is necessarily that of BNC2 expression in the tumour cells themselves.

## Discussion

The ECM consists of a complex and dynamic cross-linked meshwork of proteins that provides architectural and mechanical support to cells and initiates and regulates physical and chemical stimuli that

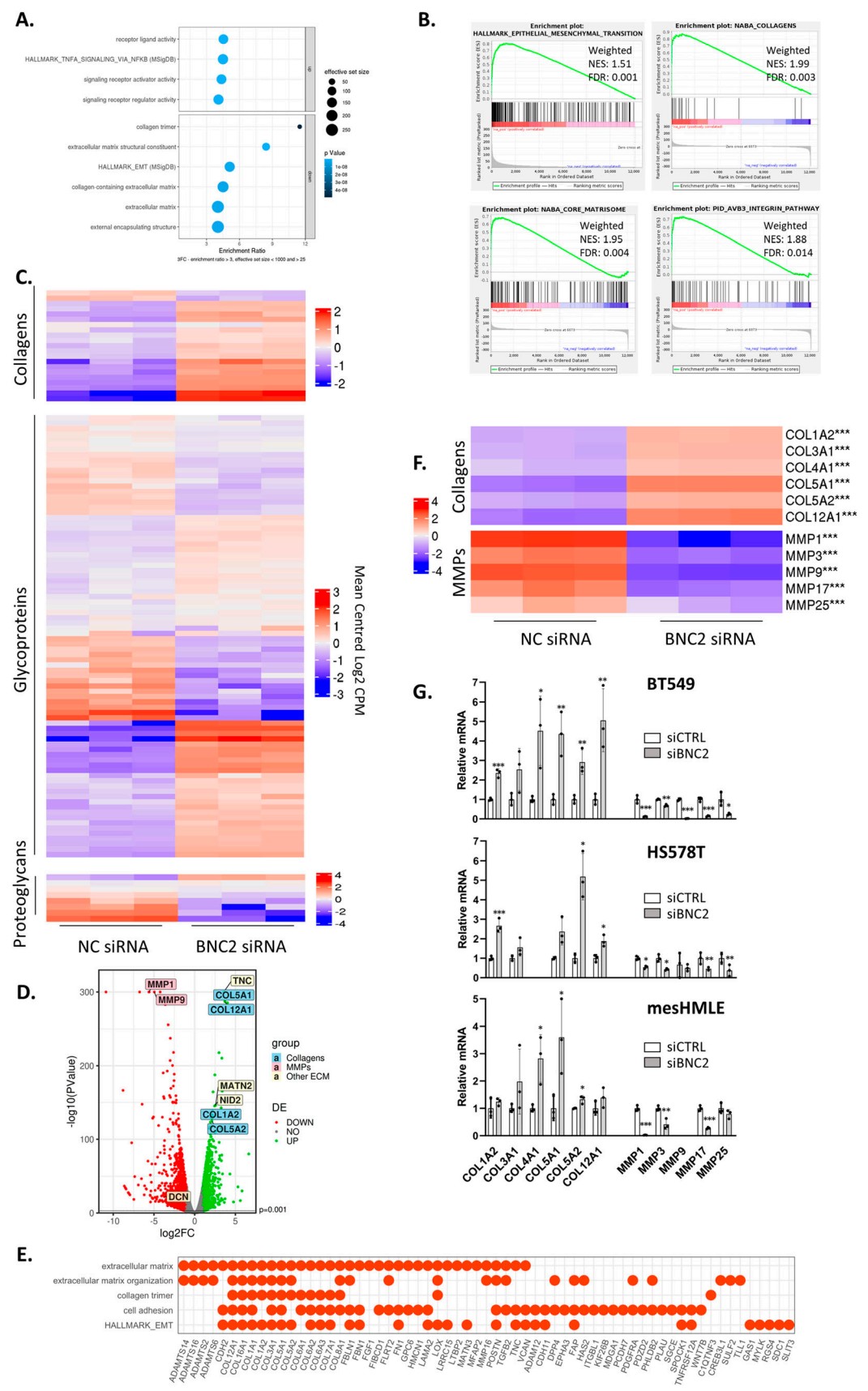

influence cell migration and invasion, proliferation, survival, differentiation, and tissue morphogenesis (60). In cancer, the ECM becomes dysregulated in composition and configuration which influences the behavior of both the cancer cells themselves and the cells that produce and modulate the ECM (such as cancer-associated and adipocyte-derived fibroblasts; CAFs and ADFs) (61, 62, 63).

In this study, we report a previously uncharacterised role for BNC2 as an important regulator of multiple, specific genes that produce, modify, and degrade the ECM. Among the most prominent of these are fibrillar collagens (type I, III, V, XII) that are causally linked to tumour cell survival and metastasis (64). In parallel with controlling expression of collagen genes, BNC2 also regulates the expression of genes whose products bind and regulate collagen proteins. For example, BNC2 knockdown down-regulates DCN, a proteoglycan that binds collagen fibrils and regulates collagen fibre formation (65). On the other hand, BNC2 knockdown up-regulates the NID2 and MATN2 glycoproteins that have roles in ECM assembly through extensive interactions with both collagenous and non-collagenous matrisomal components (66, 67).

In addition to the matrisome, BNC2 also orchestrates complex changes in the expression of multiple ECM-degrading proteases such as specific members of the MMP family that include collagenases (MMP1), gelatinases (MMP9), stromelysins (MMP3), and membrane-bound MMPs (MMP17, 25) that collectively process and degrade a broad spectrum of collagens and other ECM components. Complex regulation is also seen in the plasminogen activation (PA) system; a key mechanism of ECM remodeling. Here, the ECM-degrading protease plasmin is generated from plasminogen via the actions of urokinase (uPA) or tissue-type (tPA) plasminogen activators, encoded, respectively, by the genes PLAU and PLAT. Plasmin activation is critical for the breakdown of blood clots (fibrinolysis) and for cancer progression where PA dysregulation affects cell adhesion and migration (68). We note BNC2 knockdown results in the dramatic reduction of PLAT (whereas PLAU remains very highly expressed) and of multiple serpin genes (A1, B2, B8, E1) that are prominent uPA and tPA inhibitors (Fig S12A). This would suggest that BNC2 regulates the PA system, and thereby ECM remodeling, at multiple levels. Another example of the complex regulation orchestrated by BNC2 can be seen within the semaphorin family; secreted and membrane-associated signaling proteins that were originally identified as regulators of axon guidance, but that are now recognized to have other roles, including in metastasis (69). It is interesting that upon BNC2 knockdown, specific semaphorins and semaphorin (plexin) receptors are both up- and down-regulated, suggesting a complex regulatory system yet to be explored (Fig S12B). Lastly, we also observe that the expression of a number of pro-inflammatory cytokines is dependent

upon BNC2; most dramatically IL-1B, and also others including IL-6, IL-8, LIF, and CCL5 (Fig S12C). These are noteworthy on account of their promotion of cancer growth and metastasis, achieved via multiple mechanisms including the promotion of EMT (70, 71, 72, 73, 74) and the release of MMPs (75, 76, 77). As the expression of cytokines such as IL-1B correlates with poor prognosis across different cancer types, more advanced cancer stage and metastatic spread (78, 79, 80), the importance of BNC2 regulating these matrisome-adjacent genes also warrants further investigation.

Co-regulons have been used to suggest a putative role for BNC2 in the induction of EMT (8). GWAS that implicate BNC2-associated alleles with adolescent idiopathic scoliosis (31, 32, 33, 34, 35, 36) also implicate other ECM-associated genes and cytokines that we also find to be under BNC2 control in our breast cancer system (MATN1, MMP9, SOX9, IL-6 as examples) (35). Craniofacial developmental defects observed after BNC2 knockout (28) likely have a contribution of ECM remodeling, whereas hair follicle defects (30) suggest a developmental role in mesenchymal progenitor cells that are again consistent with a gene, that is, mesenchymally expressed in the adult with an EMT-associated function.

This study, together with a recent publication detailing the involvement of BNC2 in myofibroblasts during liver fibrosis (51), directly implicates BNC2 as a regulator of the ECM. Our findings are broadly consistent with those reported by reference 51, who showed that BNC2 is expressed in myofibroblasts, is induced during fibrosis and is co-expressed with Col1a1. Here, we too report BNC2 is most highly expressed in fibroblasts (Fig 6), is induced by TGFβ (that promotes EMT and fibrosis, Fig 1G–I) and is co-expressed with type-I collagens (Figs 3 and 8E). In contrast, however, reference 51 reports BNC2 knockdown decreased collagen expression in hepatic stellate cells, whereas we find in mammary fibroblasts (and BNC2-expressing breast cancer cells), BNC2 knockdown increases the expression of collagen (both mRNA and secreted collagen protein).

Our data from mammary cells are consistent with the same transcriptional programs driving both BNC2 and collagen expression, with BNC2 then acting upon collagen genes to limit their excessive production. The explanation for why in different contexts, BNC2 either promotes or inhibits collagen expression is unclear; however, one intriguing possibility is suggested by the model proposed by reference 51, in which YAP1 and SMAD proteins co-associate with BNC2 on the promoters of matrisomal genes to drive expression. It is interesting that YAP1 has been previously reported to convert ZEB1 from a transcriptional repressor to an activator (81). One possibility is therefore that BNC2 might play either a positive or a negative role depending upon which transcriptional co-factors are present, be that YAP1 or otherwise.

On the basis of Zn-finger domains and nuclear localization, BNC2 is widely regarded as a transcription factor and thus, an obvious

**Figure 4. Endogenous basonuclin-2 (BNC2) regulates the ECM and ECM degradation.**
**(A)** Highly enriched ontologies of genes that were responsive to BNC2 siRNA transfection in BT549 cells (up- or down-regulated >3x) as measured by RNA-seq. **(B)** Epithelial–mesenchymal transition was identified as particularly strongly enriched among BNC2-responsive genes by Gene Set Enrichment Analysis. These genes largely represent the matrisomal component of epithelial–mesenchymal transition. **(C)** Heat map of gene expression from RNA-seq showing core matrisomal genes. **(D)** Volcano plot generated from triplicate RNA-seq experiments after BNC2 knockdown. A selection of key matrisomal genes and ECM regulators are indicated. **(E)** Genes that contribute to gene ontologies that are enriched in both co-expression with BNC2 in patient weighted gene co-expression network analysis (Fig 3A) and that are regulated by BNC2 knockdown in BT549 cells (Fig 4A). **(F, G)** Select collagens and matrix metalloproteases that are responsive to BNC2 knockdown in RNA-seq (F) were measured by qRT-PCR in BT549, HS578T and mesHMLE cells (G). * indicates *P* < 0.05, **P* < 0.01, ***P* < 0.001 (unpaired *t* test) relative to control transfection.

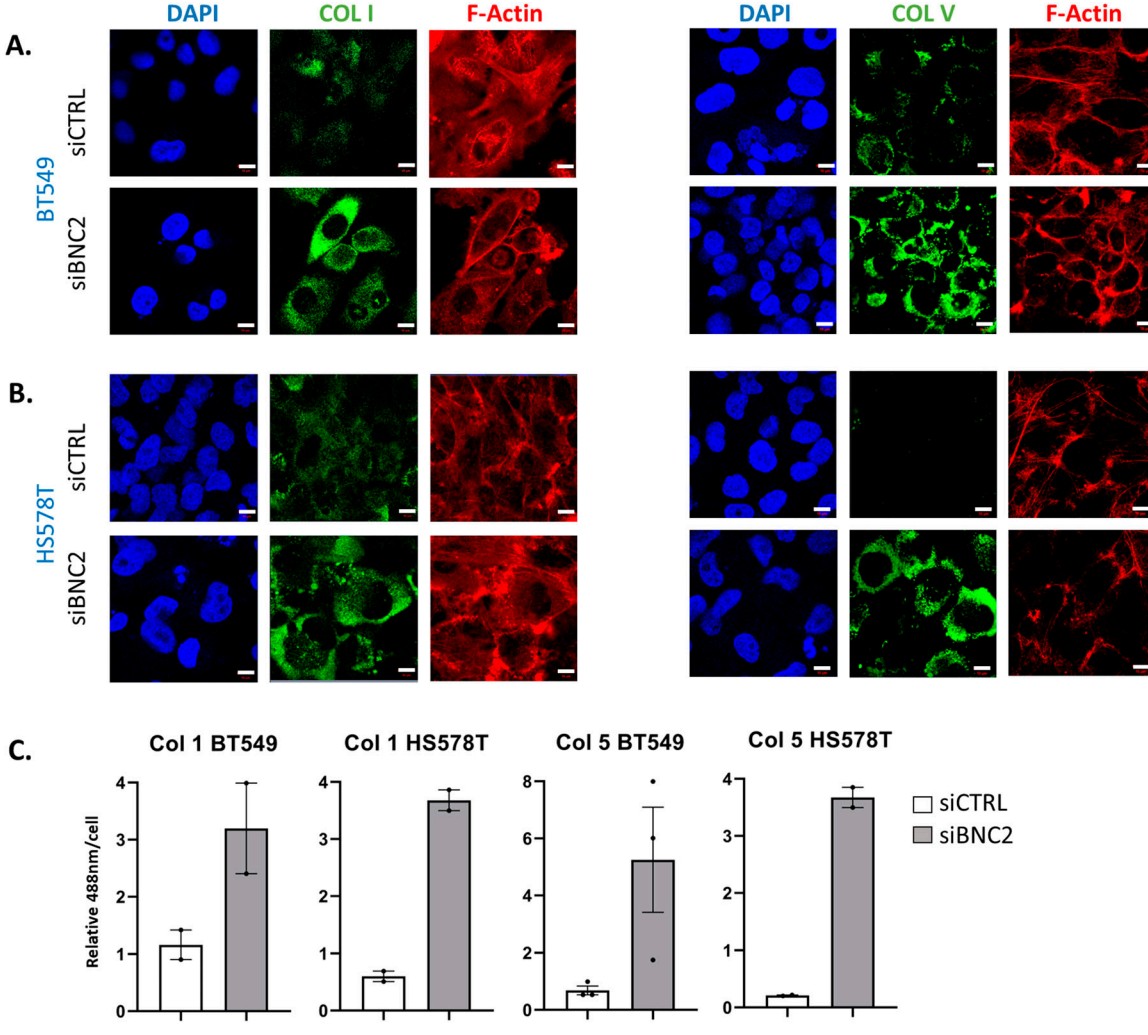

**Figure 5. Endogenous basonuclin-2 controls collagen expression.**
**(A, B)** Localisation of endogenous collagen type-I or type-V and F-actin (phalloidin) after basonuclin-2 siRNA transfection in (A) BT549 and (B) HS578T cells. Bar indicates 10 μm length. **(C)** Quantitation of collagen expression (488 nm fluorescence per cell). Individual measurements indicate average quantitation per cell from separate fields of view (~8–15 cells in each case).

mechanism to account for the effects of BNC2 is the binding of DNA and regulation of target gene transcription. To this end, two BNC2 ChIP-Seq studies have been published (10, 51) and it is noteworthy that matrisomal genes are identified as enriched in both. However, as a minor word of caution, we have not been able to demonstrate this mechanism as the same commercially available BNC2 antibody, that is, used in these studies (which exhibits questionable specificity, Fig S1), was not capable of immunoprecipitating either endogenous BNC2 or a FLAG-tagged construct in our hands. A FLAG-antibody did pulldown FLAG–BNC2; however, this failed to result in the convincing co-precipitation of DNA. As such, we are yet to fully define the complete mechanisms behind our observations though for now, direct regulation of matrisomal genes remains the leading hypothesis.

Despite extensive linkage to cancer via GWAS, a role for BNC2 as a regulator of the ECM is only now emerging. In the context of cancer, this may apply to both the cancer cells themselves and to mesenchymal stromal cells such as fibroblasts that are primarily responsible for matrix production and remodeling. The ECM as a whole, and individual ECM components such as collagens and MMPs, are widely implicated in cancer but whether in a tumour-promoting or suppressing role is complex and context dependent. Similarly, the tumour promoting or suppressing role of BNC2 appears to be similarly complex. We report seemingly contradictory findings in that BNC2 is generally expressed at lower levels in tumours compared with normal tissue (Figs 8 and S8); however, high BNC2 expression in tumours corresponds to more advanced disease, poorer prognosis, and the classification of increased types of lower-survival tumours (metaplastic and claudin-low) (Figs 8, S10, and S11). As these data were obtained from the bulk sequencing of tissue, and because BNC2 is most highly expressed in fibroblasts, one explanation is that the prognostic impact of BNC2 does not derive from BNC2 expressed in the cancer cells themselves, but as a marker of infiltrating fibroblasts. High levels of fibroblast infiltration

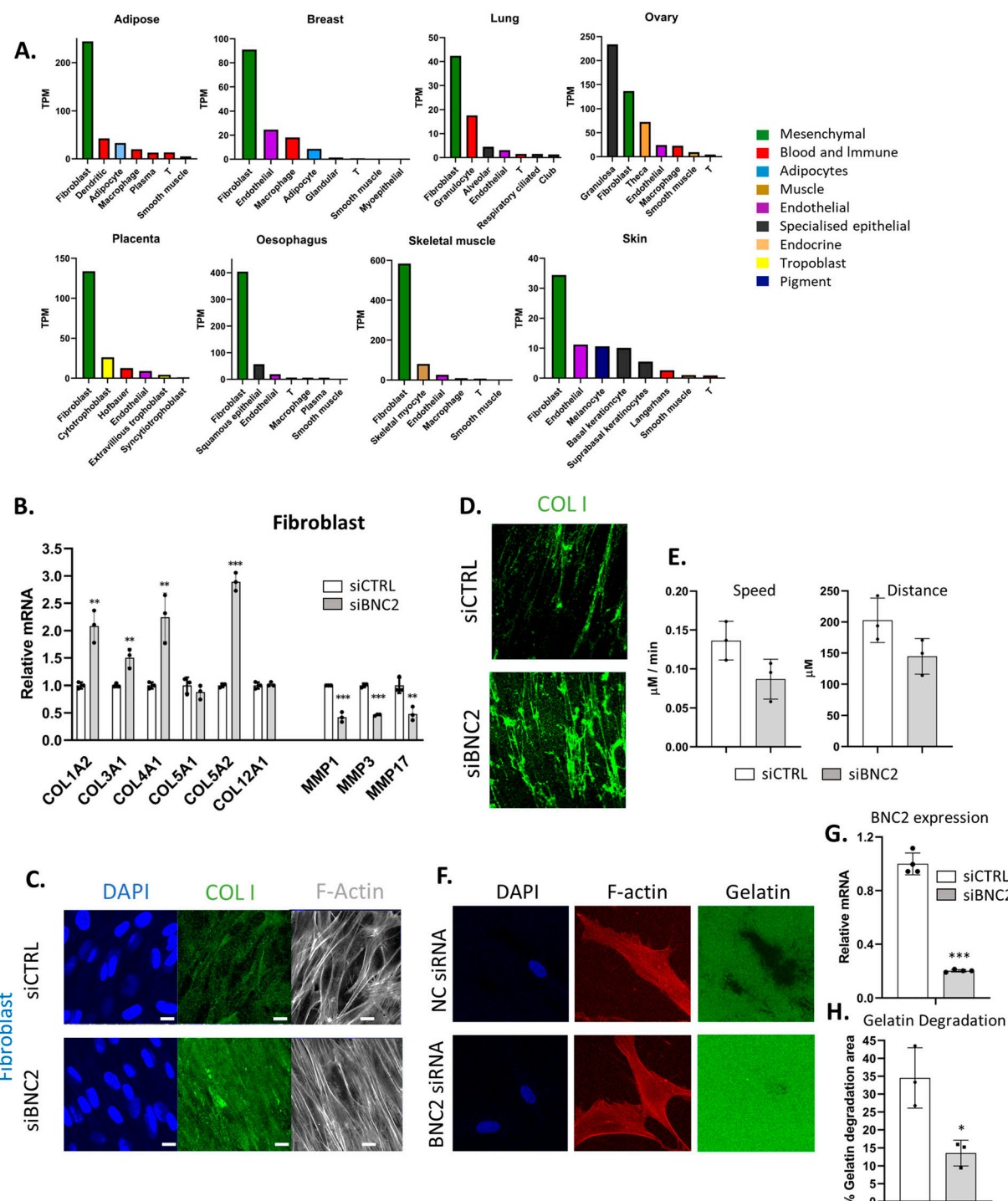

**Figure 6. Endogenous basonuclin-2 (BNC2) controls collagen deposition and matrix degradation by fibroblasts.**
**(A)** Cell-specific BNC2 expression (transcripts per million) derived from single-cell sequencing of human tissues (obtained from the Human Protein Atlas).
**(B)** Gene expression of selected collagens and matrix metalloproteases (as in Fig 4D) in response to knockdown of BNC2 in human fibroblasts (as in Fig 4D).
**(C)** Immunofluorescence analysis of endogenous collagen I and F-actin in fibroblasts in response to BNC2 siRNA. Bar indicates 10 μm length. **(D)** Immunofluorescence analysis of collagen I in cell-derived matrix prepared from fibroblasts in which control or BNC2 siRNAs were transfected. **(D, E)** Motility of GFP-labelled MDA-MB-231 breast cancer cells on a 3D matrix derived from fibroblasts (shown in (D)) monitored. Speed and total distance traveled is shown (*P*-Val = 0.074 and 0.093, respectively).
**(F)** BNC2 was knocked down in fibroblasts and digestion of fluorescent gelatin matrix was visualized. **(G)** BNC2 expression after knockdown was measured by qRT-PCR.
**(H)** Quantitation of gelatin degradation from >250 cells. * indicates *P* < 0.05, **P* < 0.01, ***P* < 0.001 (unpaired *t* test) relative to control transfection.

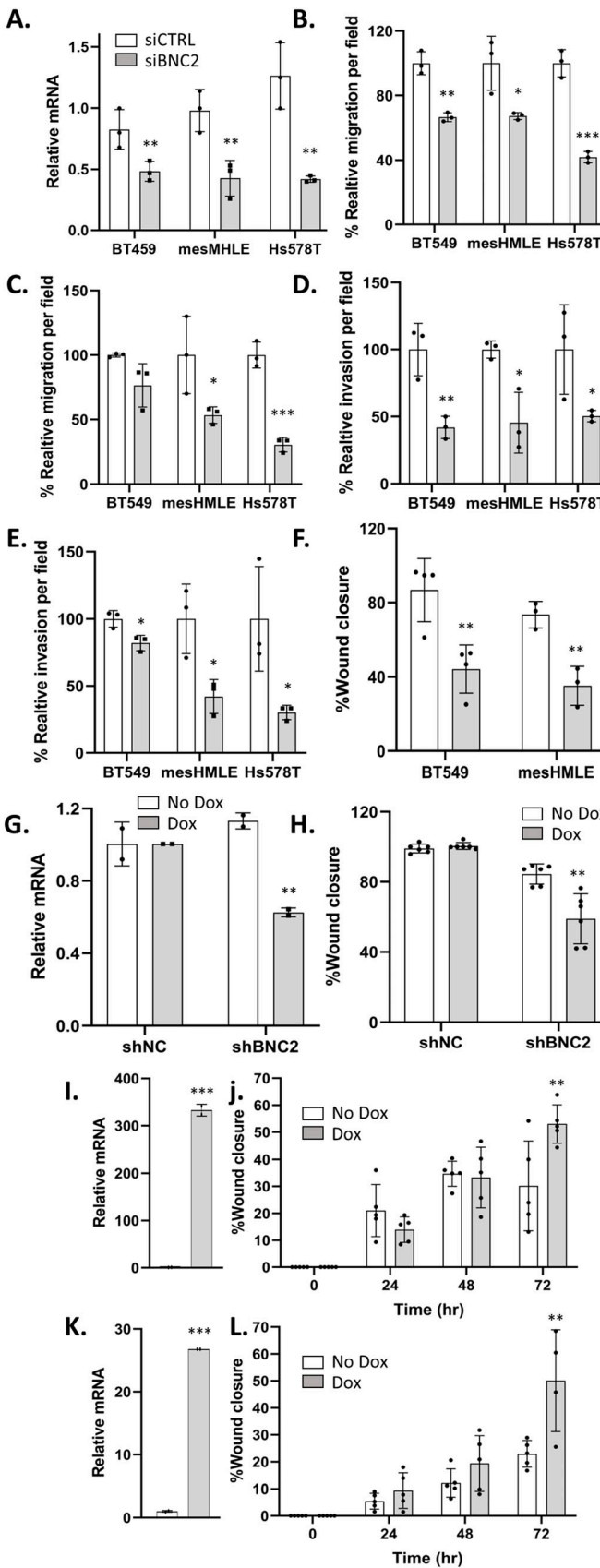

into tumours correlate with poor prognosis (82, 83). There is a possibility that BNC2 may act as a tumour suppressor in the cancer cells in which it is expressed (decreasing motility for example), but the presence of high levels of BNC2 in tumour bulk sequencing could still be indicative of fibroblast infiltration and thus, poorer outcomes. Determining whether BNC2 functions as a tumour suppressor or an oncogene is therefore a nuanced question that requires further investigation, particularly in relation to the impact on cancer development from its role in matrisome-secreting fibroblasts.

# Materials and Methods

### Cell culture and maintenance

All cell lines were maintained at 37°C with 5% $CO_2$ in media listed in Table S1. Human mammary epithelial (HMLE) cells were obtained from the Weinberg laboratory. Human mammary fibroblasts, immortalized with hTert, were a gift from Roger Reddel (Children's Medical Research Institute, CMRI, Westmead). To induce EMT, HMLE, MCF10A, and MDCK (Madin–Darby canine kidney) cells were treated with 2.5 ng/ml TGF-$\beta$ (R&D systems) for a minimum of 2 wk.

### RNA extraction and real-time PCR

Total RNA was extracted using TRIzol (Invitrogen) following the manufacturer's instructions. After RNA quantification (ND-1000; NanoDrop spectrometer), 1 $\mu$g of RNA was reverse transcribed using the QuantiTect Reverse Transcription Kit (QIAGEN). Quantitative RT–PCR was performed on a Corbett Rotor-Gene 6000 detection system (QIAGEN) using QuantiTect SYBR Green (QIAGEN). Relative gene expression levels were determined using the comparative quantification feature of Rotor-Gene software and normalized to GAPDH or ACTB. Primer sequences are listed in Table S2.

### miRNA and siRNA transfection

Cells were transfected with miRNA mimics, siRNAs, or plasmids using the Lipofectamine 2000 Transfection Reagent (Invitrogen) for plasmids or RNAiMax (Invitrogen) for siRNAs and miRNAs following manufacturer's protocols. 5–24 h post-transfection, media was replaced. The sequences of the miRNA mimics and siRNAs are listed

**Figure 7. Basonuclin-2 (BNC2) promotes cell motility and invasion.**
**(A)** Representative siRNA-mediated knockdown of BNC2 expression across immortalized breast and breast cancer cell lines. **(B, C, D, E, F)** Phenotypic effect of BNC2 knockdown on (B, C) migration, (D, E) invasion, and (F) wound healing comparing control to BNC2 siRNA–transfected cells. **(B, C, D, E, F)** siRNAs were obtained from either GenePharma (B, D, F) or QIAGEN (C, E). **(G, H)** Wound closure in BT549 cells in which BNC2 was knocked down using an inducible shRNA vector. **(I, J, K, L)** Wound closure was measured 1–3 d after the induction of full-length BNC2 by Dox in (I, J) MCF7 and (K, L) T47D cells. **(H, J, L)** Similar trends to those shown in (H, J, L) were seen with two independent clonal cell lines. * indicates $P < 0.05$, **$P < 0.01$, ***$P < 0.001$ (unpaired $t$ test) relative to control transfection, in each case making pair-wise comparisons between control and BNC2-perturbed experiments.

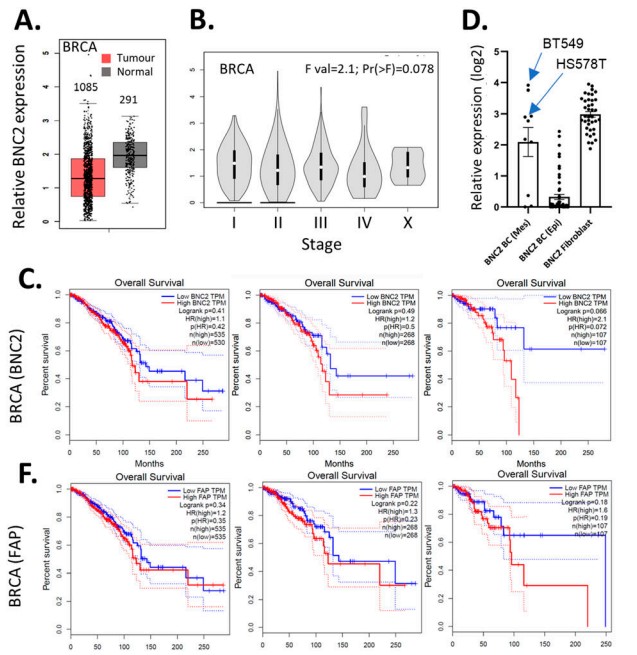

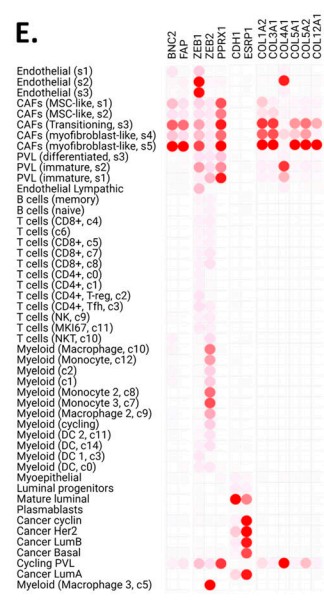

**Figure 8. Basonuclin-2 (BNC2) expression correlates with more aggressive cancer.** **(A)** BNC2 expression in the bulk sequencing of matched breast invasive carcinoma (BRCA) and normal tissue was determined using Gene Expression Profiling Interactive Analysis. **(B)** Violin plot of BNC2 expression across different breast cancer grades. **(C)** Kaplan–Meier plots indicating survival of patients with four different cancers, split by whether BNC2 high/low expression is classified as the upper versus lower half of expression (50/50) or whether only the top/bottom quartiles (75/25) or top/bottom 10th percentiles (90/10) of expression are considered. **(D)** Relative expression of BNC2 mRNA across all breast or fibroblast cell lines within the cancer cell line encyclopedia. Breast cancer lines have been sub-divided into epithelial and mesenchymal-like based upon largely mutually exclusive E-cadherin and ZEB1 expression. **(E)** Gene expression derived from single-cell sequencing of 26 breast cancer patients (58). **(C, F)** Kaplan–Meier plots correlating patient survival with FAP (fibroblast marker) expression as described in (C).

in Table S3. Final concentrations of 10 nM miR-200c mimic or 20 nM siRNA were used throughout. siRNAs were comprised by equally mixing three GenePharma or two QIAGEN siRNAs (sequences listed in Table S3) and compared with 20 nM of the respective siRNA control.

## Lentivirus production and transduction

$2 \times 10^6$ HEK293T cells were seeded in T25 flasks the day before transfection with 4 µg of the corresponding construct and viral packaging plasmids (pLP1, pLP-VSVG, and pTAT; 1 µg each) using 12 µl Lipofectamine 2000 (Invitrogen) and 500 µl Opti-MEM (Gibco). Viral supernatants were then collected after 72 h and used to transduce cells (1:4) in the presence of 4 mg/ml polybrene. Depending on the selection marker on the transfer plasmid, 1 µg/ml of puromycin or 1 mg/ml G418 was used to eradicate un-transduced cells. Cells were then grown for at least 72 h before further analyses.

## Generation of inducible 3xFlag-tagged BNC2 overexpression cell lines

Bacteria expressing *BNC2* cDNA were obtained from DNASU (clone ID# 859007). *BNC2* was then amplified by PCR and cloned into the BamH1 and Not1 restriction sites of the pENTER2B vector. 3-FLAG tags were then introduced into the BamH1 and Apa1 restriction sites. Using gateway cloning (Invitrogen), 3xFlag-*BNC2* fragments were then cloned into the pInducer20 vector. Lentivirus generation and MCF7 and T47D cell transduction was performed as described above. Flowcytometric single-cell sorting was performed using a FACSMelody Cell Sorter (BD Bioscience) to isolate single cells. To induce expression, MCF7 and T47D single clones were treated for 6 d with 0.05 and 1 µg/ml of doxycycline, respectively. qRT-PCR was then performed to screen for BNC2-expressing cells.

## Generation of stable GFP-expressing BT549 and MDA-MB-231 cells

Lentiviral generation was performed using a pLV4301-enhanced GFP transfer vector and packaging plasmid as outlined above. BT549 and MDA-MB-231 cells were then transduced with viral supernatants. After antibiotic selection (1 µg/ml puromycin), flowcytometric cell sorting was performed to purify GFP-expressing cells which were grown for a further 72 h before analyses.

## Generation of inducible shBNC2 cell lines

Specific *BNC2* shRNA and negative control shRNA were cloned into a pInducer10 vector using XhoI and EcoRI restriction sites. Lentivirus generation and BT549 cell transduction was performed as previously described. After antibiotic selection (1 µg/ml puromycin) and induction (1 µg/ml doxycycline), flow cytometry was used to isolate m-cherry–expressing cells (as it is incorporated into the pInducer vector). Single-cell sorting and qRT-PCR were then performed to identify single clones with the best BNC2 knockdown efficiency.

## Luciferase reporter assay

$4 \times 10^4$ MDA-MB-231 cells were seeded in 24-well plates and co-transfected with 10 nM scrambled or miR-200c mimics (mirVana miRNA mimic-MC11714), 100 ng firefly luciferase reporter plasmid (pGL3), and 15 ng pRL-TK Renilla plasmid into which the first 3 kb of the BNC2 3'UTR was cloned. 24 h post-transfection, cells were lysed in passive lysis buffer (Promega) and activity measured using the Dual-Luciferase Reporter Assay System (Promega) on a GloMax-Multi Detection System (Promega). Relative luciferase

activity was calculated as the ratio of Renilla to Firefly luciferase activity.

## Western analysis

Protein extracts were prepared using 1xRIPA buffer (Bio-Rad) (including protease and phosphatase inhibitors). Protein was then quantified using a BCA protein assay (Thermo Fisher Scientific). 20 $\mu g$ of lysates were separated on Bolt Bis–Tris Plus 4–12% gels (Invitrogen), transferred to nitrocellulose, and blocked with 5% skimmed milk in TBS-T. Membranes were incubated with primary antibody overnight at 4°C and detected by HRP anti-mouse or rabbit secondary antibody in 5% skimmed milk in TBS-T at RT for 1 h. Membranes were then visualised by ECL (GE Healthcare) according to the manufacturer's instructions and imaged using the ChemiDoc Touch Imaging System (Bio-Rad). All antibodies are listed in Table S4.

## Immunofluorescent labelling

Cells were seeded on either 13 mm coverslips or in fibronectin-coated chamber slides (BD Biosciences), fixed in 4% paraformaldehyde, permeabilized in 0.3% PBS–Triton X-100 for 5 min, blocked in 3% BSA/0.1% PBS-T for 20 min, and labelled with 1:500 dilution of primary antibody in 3% BSA/0.1% PBS-T overnight at 4°C, followed by the relevant goat–anti-mouse/rabbit secondary antibodies conjugated to either AlexaFluor 488 or AlexaFluor 555 in 3% BSA/0.1% PBS-T (1:500; Invitrogen). After secondary antibody incubation, F-actin labelling was performed using rhodamine phalloidin in 3% BSA/0.1% PBS-T (1:300; Invitrogen) for 30 min. Coverslips were mounted in Vectashield-containing DAPI (Vectorlabs). Confocal images were acquired using Zeiss LSM 700 or Leica SP8 confocal systems.

## Wound-healing assay

$5 \times 10^5$ cells were seeded in 24-well plates. Dox induction of *BNC2* shRNA or expression was performed for 6 d before 24 h serum starvation. A scratch was made, and live cell imaging was conducted for up to 72 h (10X objective lenses). Wound closure was calculated using ImageJ.

## Migration and invasion assay

Following two rounds of *BNC2* or control siRNA transfection, cells were seeded ($2 \times 10^5$ cells) without serum in Transwell (8.0 IM PET; Costar) or Matrigel Invasion Chambers (8.0 IM PET; BD Biocoat) covered in complete media. Cells were then incubated at 37°C for 8 and 24 h for migration and invasion assays, respectively, before fixing in 4% paraformaldehyde. Transwells and Matrigel chambers were washed extensively with PBS, permeabilized using 0.1% Triton X-100, stained with DAPI (400 ng/ml in methanol), and mounted with DAKO fluorescence mounting mediun (Agilent). Cells were counted and averaged across six fields and counted using ImageJ. All assays were performed in triplicate.

## Gelatin degradation assay

Coverslips (18 mm diameter) were prepared by subjecting to a 30% nitric acid wash (30 min) followed by 0.01% poly-L-lysine (Sigma-Aldrich) coating (15 min) and crosslinking with 0.5% glutaraldehyde (10 min, including extensive PBS washing). Gelatin coating was performed by incubating coverslips with pre-warmed 0.1% Oregon Green 488-conjugated gelatin (Invitrogen) and 0.2% porcine gelatin for 10 min to which 5 mg/ml NaBH4 (a reducing agent) was added (3 min). Coverslips were incubated in complete media for 2 h at 37°C before seeding ($2 \times 10^4$ cells/ml) and overnight incubation. Cells were then fixed with 4% paraformaldehyde in 5% sucrose (20 min at 37°C), permeabilized in 0.5% Triton X-100-PBS (5 min), and immunolabelled (1:300; AlexaFluor 647-Phalloidin). Coverslips were mounted in Vectashield-mounting media (VectorLabs). Imaging was performed using Zeiss LSM 700 and Leica SP8 confocal microscopes. Quantitation of gelatin degradation was performed counting a minimum of 250 cells per variable. The total area of gelatin degradation was calculated using ImageJ. All assays were performed in triplicate.

## CDM

Using a similar protocol to that previously published (84). Briefly, 18 mm sterilized coverslips were incubated in 0.2% gelatin (1 h at 37°C), washed with PBS, and cross-linked using 1% glutaraldehyde (30 min at RT). Remaining glutaraldehyde was quenched using 1 M glycine (20 min at RT) and washed again with PBS. Coated coverslips were incubated with complete DMEM for 1 h at 37°C before seeding immortalized fibroblasts at $1 \times 10^5$ cells per coverslip, and incubated overnight at 37°C before changing to medium supplemented with 50 $\mu g$/ml ascorbic acid for 7 d. Medium was changed daily. Cells were lysed with pre-warmed extraction buffer (20 mM NH$_4$OH and 0.5% Triton X-100 in PBS) for 5 min until cell denudation was observed. Residual DNA was removed by incubation with 10 $\mu g$/ml DNase I (37°C for 30 min) followed by 2x PBS washes. The CDM were either used for immunofluorescence analysis of collagen I and MMP1 or seeded with breast cancer cell lines followed by live cell imaging for 48 h using an IN Cell Analyzer 2200 (Cytiva) then cell movement was analyzed by ImageJ TrackMate plugin (85).

## Flow cytometry and single-cell sorting

Cells were washed in PBS, trypsinized, washed twice in PBS containing 0.01 M EDTA, and centrifuged at 350*g* for 5 min followed by resuspension of the cell pellet in sorting buffer (ice cold 1xPBS, 5 mM EDTA, 1% FCS, and 25 mM HEPES; pH 7.0). Cells were then filtered using 30 $\mu m$ sterile filters (BD Biosciences) and sorted based on GFP or m-Cherry intensity compared with parental cells using a BD FACSMelody Cell Sorter. Where single clones were needed, pooled cells were sorted into single cells in 96-well plates and grown in conditioned media before expanding into larger flasks.

## RNA sequencing

RNA was prepared from cells using TRIzol according to the manufacturer's instructions. RNA quality was then checked using an Agilent Bioanalyzer (all RIN values > 7) and concentration confirmed using a Qubit HS RNA kit. Poly(A) RNA was then isolated using NEBNext Poly(A) mRNA Magnetic Isolation Module columns and the library prepared with KAPA RNA HyperPrep kits according to the manufacturer's instruction. Library sizes and yields were confirmed with the Agilent Bioanalyzer and Qubit and were diluted to 4 nM stocks. Libraries were pooled in equimolar ratios and sequenced using a single-end 75 cycle high output kit on an Illumina Next-Seq500 at the ACRF Cancer Genomics Facility. Three biological replicates were prepared with and without knockdown of BNC2 in BT549 cells. BNC2 knockdown was achieved using GenePharma siRNAs, with RNA prepared 6 d after siRNA transfection (specifically chosen to allow sufficient time for the loss of BNC2 to permeate across the gene expression network).

### Sequencing read quality control and mapping

The FASTQ files were analyzed and quality-checked using FASTQC (https://www.bioinformatics.babraham.ac.uk/projects/fastqc/). Reads were mapped against the human reference genome (build hg19) using the iGenomes UCSC hg19 gene annotations using the STAR (v2.6.1d) spliced alignment algorithm (86) with parameters—twopassMode basic and quantMode GeneCounts and otherwise default parameters. Alignments were visualized and interrogated using the Integrative Genomics Viewer v2.12.2 (87).

### Differential expression analysis

Differential gene expression was performed using EdgeR (88) glmFit and glmLRT methods to fit negative binomial generalized log-linear models to the gene-level read counts produced by STAR. Genes with a $Q$-value of ≤ 0.05 were considered significant.

### Breast cancer gene correlation network analysis

Weighted gene correlation network analysis was performed using the R package WGCNA (89). A signed network was constructed from gene expression data from TCGA breast cancer cohort (BRCA) for primary tumours only (n = 1,094) in counts per million (CPM) format. Using only genes on autosomal chromosomes, highly expressed genes were selected by requiring expression to be at least 10 cpm in at least 20 samples. A pseudocount of one as added to the raw counts before $\log_2$ transformation whereupon we selected the 5,000 genes with the largest range between the 90th and 10th percentiles of expression. The distance matrix was created using distOptions = "method = euclidean" and a Z.k threshold of −3 was used which resulted in identifying and discarding three samples as outliers. The data were raised to a soft power of eight and used to produce a tree from the topological dissimilarity matrix using method = "average." To select gene clusters, we performed dynamic branch cutting (method = "hybrid," minClusterSize = 20, deepSplitParam = 2) and merged closely related modules using the mergeClose function (cutHeight = 0.15). Module eigengenes were determined and correlated with traits,

and the statistical significance of this was evaluated using the cor (use = "p") and the corPvalueStudent functions. Module membership, a generalised version of intramodular connectivity, was calculated for each gene and module eigengene combination, defined as the correlation between the two. The resulting network was exported using the exportNetworktoCytoscape function (with threshold = 0.1) visualised in Gephi (90).

### Gene ontology analysis

The gene ontology enrichment in WGCNA gene modules and BNC2 knockdown samples was characterised using the anRichment package with the entire GO collection for "human" and Hallmark collection from MSigDB (v7.4, organism = "human," "MSigDB H: Hallmark gene sets"). GO enrichment was calculated using a threshold of "5 × 10$^{-2}$" for thresholdType = "Bonferroni." We discarded gene sets larger than 1,000 or smaller than 25 genes.

### Statistics and reproducibility

Statistical analysis was carried out using GraphPad Prism 8 software, and data are represented as the mean with SD. Significance was measured by two-tailed unpaired $t$ tests, comparing BNC2 manipulated and control samples for all qRT-PCR, luciferase, and migration/invasion assays presented. Significant $P$-values are indicated by $*P < 0.05$, $**P < 0.01$, $***P < 0.001$ relative to control transfection as stated in figure legends. Pearson correlation of BNC2 expression with epithelial/mesenchymal gene score (Fig 1A–C) and other genes (Fig S2A and B) is shown. Statistical significance of enriched gene ontologies are calculated using the bioinformatic packages described.

# Data Availability

The RNA sequencing data from this publication have been deposited to the NCBI's Gene Expression Omnibus and are accessible through GEO Series accession number GSE221228.

# Supplementary Information

# Acknowledgements

The authors would like to acknowledge the following funding sources: ARC Future Fellowship (FT190100544), the Hospital Research Foundation (2022-S-DTFA-007), and the Royal Adelaide Hospital Research Foundation Florey Fellowship.

### Author Contributions

A Orang, BK Dredge, CY Liu, L Sourdin, and R Lumb: investigation and methodology.

JM Bracken, C-H Chen, and ST Boyle: formal analysis and investigation.
HJ Whitfield, MJ Davis, MS Samuel, PA Gregory, and Y Khew-Goodall: formal analysis.
GJ Goodall: conceptualization, formal analysis, investigation, and methodology.
KA Pillman: conceptualization, formal analysis, supervision, investigation, methodology, and writing—original draft, review, and editing.
CP Bracken: conceptualization, formal analysis, supervision, and writing—original draft, review, and editing.

## Conflict of Interest Statement

The authors declare that they have no conflict of interest.

8. da Silveira WA, Palma PVB, Sicchieri RD, Villacis RAR, Mandarano LRM, Oliveira TMG, Antonio HMR, Andrade JM, Muglia VF, Rogatto SR, et al (2017) Transcription factor networks derived from breast cancer stem cells control the immune response in the basal subtype. *Sci Rep* 7: 2851. doi:10.1038/s41598-017-02761-6

12. Vigorito E, Kuchenbaecker KB, Beesley J, Adlard J, Agnarsson BA, Andrulis IL, Arun BK, Barjhoux L, Belotti M, Benitez J, et al (2016) Fine-scale mapping at 9p22.2 identifies candidate causal variants that modify ovarian cancer risk in BRCA1 and BRCA2 mutation carriers. *PLoS One* 11: e0158801. doi:10.1371/journal.pone.0158801

with anti-cancerous properties in a lung cancer model. *Cancer Cell Int* 17: 18. doi:10.1186/s12935-017-0394-x

28. Vanhoutteghem A, Maciejewski-Duval A, Bouche C, Delhomme B, Herve F, Daubigney F, Soubigou G, Araki M, Araki K, Yamamura K-I, et al (2009) Basonuclin 2 has a function in the multiplication of embryonic craniofacial mesenchymal cells and is orthologous to disco proteins. *Proc Natl Acad Sci U S A* 106: 14432–14437. doi:10.1073/pnas.0905840106

29. Yamamoto S, Uchida Y, Ohtani T, Nozaki E, Yin C, Gotoh Y, Yakushiji-Kaminatsui N, Higashiyama T, Suzuki T, Takemoto T, et al (2019) Hoxa13 regulates expression of common Hox target genes involved in cartilage development to coordinate the expansion of the autopodal anlage. *Dev Growth Differ* 61: 228–251. doi:10.1111/dgd.12601

30. Vanhoutteghem A, Delhomme B, Herve F, Nondier I, Petit JM, Araki M, Araki K, Djian P (2016) The importance of basonuclin 2 in adult mice and its relation to basonuclin 1. *Mech Dev* 140: 53–73. doi:10.1016/j.mod.2016.02.002

31. Makki N, Zhao J, Liu Z, Eckalbar WL, Ushiki A, Khanshour AM, Wu J, Rios J, Gray RS, Wise CA, et al (2021) Genomic characterization of the adolescent idiopathic scoliosis-associated transcriptome and regulome. *Hum Mol Genet* 29: 3606–3615. doi:10.1093/hmg/ddaa242

32. Man GC, Tang NL, Chan TF, Lam TP, Li JW, Ng BK, Zhu Z, Qiu Y, Cheng JCY (2019) Replication study for the association of GWAS-associated loci with adolescent idiopathic scoliosis susceptibility and curve progression in a Chinese population. *Spine (Phila Pa 1976)* 44: 464–471. doi:10.1097/BRS.0000000000002866

33. Ogura Y, Kou I, Miura S, Takahashi A, Xu L, Takeda K, Takahashi Y, Kono K, Kawakami N, Uno K, et al (2015) A functional SNP in BNC2 is associated with adolescent idiopathic scoliosis. *Am J Hum Genet* 97: 337–342. doi:10.1016/j.ajhg.2015.06.012

34. Ogura Y, Takeda K, Kou I, Khanshour A, Grauers A, Zhou H, Liu G, Fan YH, Zhou T, Wu Z, et al (2018) An international meta-analysis confirms the association of BNC2 with adolescent idiopathic scoliosis. *Sci Rep* 8: 4730. doi:10.1038/s41598-018-22552-x

35. Wang W, Chen T, Liu Y, Wang S, Yang N, Luo M (2022) Predictive value of single-nucleotide polymorphisms in curve progression of adolescent idiopathic scoliosis. *Eur Spine J* 31: 2311–2325. doi:10.1007/s00586-022-07213-y

36. Xu L, Wu Z, Xia C, Tang N, Cheng JCY, Qiu Y, Zhu Z (2019) A genetic predictive model estimating the risk of developing adolescent idiopathic scoliosis. *Curr Genomics* 20: 246–251. doi:10.2174/1389202920666190730132411

37. Kolvenbach CM, Dworschak GC, Frese S, Japp AS, Schuster P, Wenzlitschke N, Yilmaz Ö, Lopes FM, Pryalukhin A, Schierbaum L, et al (2019) Rare variants in BNC2 are implicated in autosomal-dominant congenital lower urinary-tract obstruction. *Am J Hum Genet* 104: 994–1006. doi:10.1016/j.ajhg.2019.03.023

38. Bhoj EJ, Ramos P, Baker LA, Garg V, Cost N, Nordenskjold A, Elder FF, Bleyl SB, Bowles NE, Arrington CB, et al (2011) Human balanced translocation and mouse gene inactivation implicate basonuclin 2 in distal urethral development. *Eur J Hum Genet* 19: 540–546. doi:10.1038/ejhg.2010.245

39. Kon M, Suzuki E, Dung VC, Hasegawa Y, Mitsui T, Muroya K, Ueoka K, Igarashi N, Nagasaki K, Oto Y, et al (2015) Molecular basis of non-syndromic hypospadias: Systematic mutation screening and genome-wide copy-number analysis of 62 patients. *Hum Reprod* 30: 499–506. doi:10.1093/humrep/deu364

40. Lang MR, Patterson LB, Gordon TN, Johnson SL, Parichy DM (2009) Basonuclin-2 requirements for zebrafish adult pigment pattern development and female fertility. *PLoS Genet* 5: e1000744. doi:10.1371/journal.pgen.1000744

41. Patterson LB, Parichy DM (2013) Interactions with iridophores and the tissue environment required for patterning melanophores and xanthophores during zebrafish adult pigment stripe formation. *PLoS Genet* 9: e1003561. doi:10.1371/journal.pgen.1003561

42. Endo C, Johnson TA, Morino R, Nakazono K, Kamitsuji S, Akita M, Kawajiri M, Yamasaki T, Kami A, Hoshi Y, et al (2018) Genome-wide association study in Japanese females identifies fifteen novel skin-related trait associations. *Sci Rep* 8: 8974. doi:10.1038/s41598-018-27145-2

43. Eriksson N, Macpherson JM, Tung JY, Hon LS, Naughton B, Saxonov S, Avey L, Wojcicki A, Pe'er I, Mountain J (2010) Web-based, participant-driven studies yield novel genetic associations for common traits. *PLoS Genet* 6: e1000993. doi:10.1371/journal.pgen.1000993

44. Gao W, Tan J, Huls A, Ding A, Liu Y, Matsui MS, Vierkötter A, Krutmann J, Schikowski T, Jin L, et al (2017) Genetic variants associated with skin aging in the Chinese Han population. *J Dermatol Sci* 86: 21–29. doi:10.1016/j.jdermsci.2016.12.017

45. Hernando B, Ibanez MV, Deserio-Cuesta JA, Soria-Navarro R, Vilar-Sastre I, Martinez-Cadenas C (2018) Genetic determinants of freckle occurrence in the Spanish population: Towards ephelides prediction from human DNA samples. *Forensic Sci Int Genet* 33: 38–47. doi:10.1016/j.fsigen.2017.11.013

46. Jacobs LC, Hamer MA, Gunn DA, Deelen J, Lall JS, van Heemst D, Uh HW, Hofman A, Uitterlinden AG, Griffiths CEM, et al (2015) A genome-wide association study identifies the skin color genes IRF4, MC1R, ASIP, and BNC2 influencing facial pigmented spots. *J Invest Dermatol* 135: 1735–1742. doi:10.1038/jid.2015.62

47. Jacobs LC, Wollstein A, Lao O, Hofman A, Klaver CC, Uitterlinden AG, Nijsten T, Kayser M, Liu F (2013) Comprehensive candidate gene study highlights UGT1A and BNC2 as new genes determining continuous skin color variation in Europeans. *Hum Genet* 132: 147–158. doi:10.1007/s00439-012-1232-9

48. Kukla-Bartoszek M, Pośpiech E, Woźniak A, Boroń M, Karłowska-Pik J, Teisseyre P, Zubańska M, Bronikowska A, Grzybowski T, Płoski R, et al (2019) DNA-based predictive models for the presence of freckles. *Forensic Sci Int Genet* 42: 252–259. doi:10.1016/j.fsigen.2019.07.012

49. Seo JY, You SW, Shin JG, Kim Y, Park SG, Won HH, Kang NG (2022) GWAS identifies multiple genetic loci for skin color in Korean women. *J Invest Dermatol* 142: 1077–1084. doi:10.1016/j.jid.2021.08.440

50. Visser M, Palstra RJ, Kayser M (2014) Human skin color is influenced by an intergenic DNA polymorphism regulating transcription of the nearby BNC2 pigmentation gene. *Hum Mol Genet* 23: 5750–5762. doi:10.1093/hmg/ddu289

51. Bobowski-Gerard M, Boulet C, Zummo FP, Dubois-Chevalier J, Gheeraert C, Bou Saleh M, Strub JM, Farce A, Ploton M, Guille L, et al (2022) Functional genomics uncovers the transcription factor BNC2 as required for myofibroblastic activation in fibrosis. *Nat Commun* 13: 5324. doi:10.1038/s41467-022-33063-9

52. Cursons J, Leuchowius KJ, Waltham M, Tomaskovic-Crook E, Foroutan M, Bracken CP, Redfern A, Crampin EJ, Street I, Davis MJ, et al (2015) Stimulus-dependent differences in signalling regulate epithelial-mesenchymal plasticity and change the effects of drugs in breast cancer cell lines. *Cell Commun Signal* 13: 26. doi:10.1186/s12964-015-0106-x

53. Agarwal V, Bell GW, Nam JW, Bartel DP (2015) Predicting effective microRNA target sites in mammalian mRNAs. *Elife* 4: e05005. doi:10.7554/eLife.05005

54. Migault M, Sapkota S, Bracken CP (2022) Transcriptional and post-transcriptional control of epithelial-mesenchymal plasticity: Why so many regulators? *Cell Mol Life Sci* 79: 182. doi:10.1007/s00018-022-04199-0

55. Tsherniak A, Vazquez F, Montgomery PG, Weir BA, Kryukov G, Cowley GS, Gill S, Harrington WF, Pantel S, Krill-Burger JM, et al (2017) Defining a cancer dependency map. *Cell* 170: 564–576.e16. doi:10.1016/j.cell.2017.06.010

56. Liu T, Zhou L, Li D, Andl T, Zhang Y (2019) Cancer-associated fibroblasts build and secure the tumor microenvironment. *Front Cell Dev Biol* 7: 60. doi:10.3389/fcell.2019.00060

57. Boyle ST, Johan MZ, Samuel MS (2020) Tumour-directed microenvironment remodelling at a glance. *J Cell Sci* 133: jcs247783. doi:10.1242/jcs.247783

58. Wu SZ, Al-Eryani G, Roden DL, Junankar S, Harvey K, Andersson A, Thennavan A, Wang C, Torpy JR, Bartonicek N, et al (2021) A single-cell and spatially resolved atlas of human breast cancers. *Nat Genet* 53: 1334–1347. doi:10.1038/s41588-021-00911-1

59. Plikus MV, Wang X, Sinha S, Forte E, Thompson SM, Herzog EL, Driskell RR, Rosenthal N, Biernaskie J, Horsley V (2021) Fibroblasts: Origins, definitions, and functions in health and disease. *Cell* 184: 3852–3872. doi:10.1016/j.cell.2021.06.024

60. Kolesnikoff N, Chen CH, Samuel MS (2022) Interrelationships between the extracellular matrix and the immune microenvironment that govern epithelial tumour progression. *Clin Sci (Lond)* 136: 361–377. doi:10.1042/CS20210679

61. Cox TR (2021) The matrix in cancer. *Nat Rev Cancer* 21: 217–238. doi:10.1038/s41568-020-00329-7

62. Lepucki A, Orlinska K, Mielczarek-Palacz A, Kabut J, Olczyk P, Komosinska-Vassev K (2022) The role of extracellular matrix proteins in breast cancer. *J Clin Med* 11: 1250. doi:10.3390/jcm11051250

63. Shao X, Taha IN, Clauser KR, Gao YT, Naba A (2020) MatrisomeDB: The ECM-protein knowledge database. *Nucleic Acids Res* 48: D1136–D1144. doi:10.1093/nar/gkz849

64. Cox TR, Erler JT (2014) Molecular pathways: Connecting fibrosis and solid tumor metastasis. *Clin Cancer Res* 20: 3637–3643. doi:10.1158/1078-0432.CCR-13-1059

65. Danielson KG, Baribault H, Holmes DF, Graham H, Kadler KE, Iozzo RV (1997) Targeted disruption of decorin leads to abnormal collagen fibril morphology and skin fragility. *J Cell Biol* 136: 729–743. doi:10.1083/jcb.136.3.729

66. Kohfeldt E, Sasaki T, Gohring W, Timpl R (1998) Nidogen-2: A new basement membrane protein with diverse binding properties. *J Mol Biol* 282: 99–109. doi:10.1006/jmbi.1998.2004

67. Piecha D, Wiberg C, Morgelin M, Reinhardt DP, Deak F, Maurer P, Paulsson M (2002) Matrilin-2 interacts with itself and with other extracellular matrix proteins. *Biochem J* 367: 715–721. doi:10.1042/BJ20021069

68. Heissig B, Salama Y, Osada T, Okumura K, Hattori K (2021) The multifaceted role of plasminogen in cancer. *Int J Mol Sci* 22: 2304. doi:10.3390/ijms22052304

69. Hu S, Zhu L (2018) Semaphorins and their receptors: From axonal guidance to atherosclerosis. *Front Physiol* 9: 1236. doi:10.3389/fphys.2018.01236

70. Browning L, Patel MR, Horvath EB, Tawara K, Jorcyk CL (2018) IL-6 and ovarian cancer: Inflammatory cytokines in promotion of metastasis. *Cancer Manag Res* 10: 6685–6693. doi:10.2147/CMAR.S179189

71. Fernando RI, Castillo MD, Litzinger M, Hamilton DH, Palena C (2011) IL-8 signaling plays a critical role in the epithelial-mesenchymal transition of human carcinoma cells. *Cancer Res* 71: 5296–5306. doi:10.1158/0008-5472.CAN-11-0156

72. Li R, Ong SL, Tran LM, Jing Z, Liu B, Park SJ, Huang ZL, Walser TC, Heinrich EL, Lee G, et al (2020) Chronic IL-1β-induced inflammation regulates epithelial-to-mesenchymal transition memory phenotypes via epigenetic modifications in non-small cell lung cancer. *Sci Rep* 10: 377. doi:10.1038/s41598-019-57285-y

73. Long H, Xiang T, Qi W, Huang J, Chen J, He L, Liang Z, Guo B, Li Y, Xie R, et al (2015) CD133+ ovarian cancer stem-like cells promote non-stem cancer cell metastasis via CCL5 induced epithelial-mesenchymal transition. *Oncotarget* 6: 5846–5859. doi:10.18632/oncotarget.3462

74. Yue X, Zhao Y, Zhang C, Li J, Liu Z, Liu J, Hu W (2016) Leukemia inhibitory factor promotes EMT through STAT3-dependent miR-21 induction. *Oncotarget* 7: 3777–3790. doi:10.18632/oncotarget.6756

75. Brown RD, Jones GM, Laird RE, Hudson P, Long CS (2007) Cytokines regulate matrix metalloproteinases and migration in cardiac fibroblasts. *Biochem Biophys Res Commun* 362: 200–205. doi:10.1016/j.bbrc.2007.08.003

76. Morein D, Erlichman N, Ben-Baruch A (2020) Beyond cell motility: The expanding roles of chemokines and their receptors in malignancy. *Front Immunol* 11: 952. doi:10.3389/fimmu.2020.00952

77. Tasaki K, Shintani Y, Saotome T, Andoh A, Fujiyama Y, Hozawa S, Bamba T (2003) Pro-inflammatory cytokine-induced matrix metalloproteinase-1 (MMP-1) secretion in human pancreatic periacinar myofibroblasts. *Pancreatology* 3: 414–421. doi:10.1159/000073889

78. Tulotta C, Lefley DV, Freeman K, Gregory WM, Hanby AM, Heath PR, Nutter F, Wilkinson JM, Spicer-Hadlington AR, Liu X, et al (2019) Endogenous production of IL1B by breast cancer cells drives metastasis and colonization of the bone microenvironment. *Clin Cancer Res* 25: 2769–2782. doi:10.1158/1078-0432.CCR-18-2202

79. Wu TC, Xu K, Martinek J, Young RR, Banchereau R, George J, Turner J, Kim KI, Zurawski S, Wang X, et al (2018) IL1 receptor antagonist controls transcriptional signature of inflammation in patients with metastatic breast cancer. *Cancer Res* 78: 5243–5258. doi:10.1158/0008-5472.CAN-18-0413

80. Zhou J, Tulotta C, Ottewell PD (2022) IL-1β in breast cancer bone metastasis. *Expert Rev Mol Med* 24: e11. doi:10.1017/erm.2022.4

81. Lehmann W, Mossmann D, Kleemann J, Mock K, Meisinger C, Brummer T, Herr R, Brabletz S, Stemmler MP, Brabletz T (2016) ZEB1 turns into a transcriptional activator by interacting with YAP1 in aggressive cancer types. *Nat Commun* 7: 10498. doi:10.1038/ncomms10498

82. Mak TK, Li X, Huang H, Wu K, Huang Z, He Y, Zhang C (2022) The cancer-associated fibroblast-related signature predicts prognosis and indicates immune microenvironment infiltration in gastric cancer. *Front Immunol* 13: 951214. doi:10.3389/fimmu.2022.951214

83. Mhaidly R, Mechta-Grigoriou F (2021) Role of cancer-associated fibroblast subpopulations in immune infiltration, as a new means of treatment in cancer. *Immunol Rev* 302: 259–272. doi:10.1111/imr.12978

84. Kaukonen R, Jacquemet G, Hamidi H, Ivaska J (2017) Cell-derived matrices for studying cell proliferation and directional migration in a complex 3D microenvironment. *Nat Protoc* 12: 2376–2390. doi:10.1038/nprot.2017.107

85. Ershov D, Phan MS, Pylvanainen JW, Rigaud SU, Le Blanc L, Charles-Orszag A, Conway JRW, Laine RF, Roy NH, Bonazzi D, et al (2022) TrackMate 7: Integrating state-of-the-art segmentation algorithms into tracking pipelines. *Nat Methods* 19: 829–832. doi:10.1038/s41592-022-01507-1

86. Dobin A, Davis CA, Schlesinger F, Drenkow J, Zaleski C, Jha S, Batut P, Chaisson M, Gingeras TR (2013) STAR: Ultrafast universal RNA-seq aligner. *Bioinformatics* 29: 15–21. doi:10.1093/bioinformatics/bts635

87. Thorvaldsdottir H, Robinson JT, Mesirov JP (2013) Integrative genomics viewer (IGV): High-performance genomics data visualization and exploration. *Brief Bioinform* 14: 178–192. doi:10.1093/bib/bbs017

88. McCarthy DJ, Chen Y, Smyth GK (2012) Differential expression analysis of multifactor RNA-Seq experiments with respect to biological variation. *Nucleic Acids Res* 40: 4288–4297. doi:10.1093/nar/gks042

89. Langfelder P, Horvath S (2008) WGCNA: An R package for weighted correlation network analysis. *BMC Bioinformatics* 9: 559. doi:10.1186/1471-2105-9-559

90. Bastian M, Heymann S, Mesirov JP (2009) Gephi: An open source software for exploring and manipulating networks. International AAAI Conference on Weblogs and Social Media, Vol. 3, pp 361–362. San Jose, CA: ICWSM 2009.

