## [Reviewer comments · Life Science Alliance]

Life Science Alliance

Basonuclin-2 regulates extracellular matrix production and degradation

Ayla Orang, B Dredge, Chi Liu, Julie Bracken, Chun Chen, Laura Sourdin, Holly Whitfield, Rachael Lumb, Sarah Boyle, Melissa Davis, Michael Samuel, Philip Gregory, Yeesim Goodall, Gregory Goodall, Katherine Pillman, and Cameron Bracken.

DOI: <https://doi.org/10.26508/lsa.202301984>

Corresponding author(s): Cameron Bracken, University of South Australia and Katherine Pillman,

Review Timeline:	Submission Date:	2023-02-09
	Editorial Decision:	2023-03-15
	Revision Received:	2023-06-14
	Editorial Decision:	2023-07-10
	Revision Received:	2023-07-16
	Accepted:	2023-07-20

Scientific Editor: Novella Guidi

Transaction Report:

March 15, 2023

Re: Life Science Alliance manuscript #LSA-2023-01984-T

Dr. Cameron Bracken
University of South Australia
Centre for Cancer Biology
Bradley Building, Frome Rd
Adelaide 5000
Australia

Dear Dr. Bracken,

Thank you for submitting your manuscript entitled "Basonuclin-2 regulates extracellular matrix production and degradation" to Life Science Alliance. The manuscript was assessed by expert reviewers, whose comments are appended to this letter. We invite you to submit a revised manuscript addressing the Reviewer comments.

Thank you for this interesting contribution to Life Science Alliance. We are looking forward to receiving your revised manuscript.

Sincerely,

B. MANUSCRIPT ORGANIZATION AND FORMATTING:

Reviewer #1 (Comments to the Authors (Required)):

Orang et al report that BNC2 controls ECM composition and degradation by breast cancer cells and CAFs. The authors further identified that BNC2 modulates cancer cell motility and invasion properties in line with BNC2 expression being linked to poor patient prognosis.

Many data were obtained using stable cell-lines. However, if data were made robust through the use of different clones is not stated.

Greater details regarding shBNC2 used are needed.

Several siRNA for BNC2 are reported in table S3 but how they were used in the different experiments and whether consistent data were obtained is not described.

The authors need to check that on results obtained in different assays such as wound healing and migration assays are not biased by an effect of BNC2 on cell proliferation.

The authors state that BNC2 is not a general promoter of EMT when analyzing individual genes (p12) but latter define that BNC2 is a core component of an EMT expression module when mining RANseq data (p13). These data and conclusions should be reconciled. How des data from fig2 and 4 compare?

Regulation of ECM genes including collagen seems at odds with previous reports (e.g. Ref 51) but this is not discussed. Details regarding the fibroblasts used in this study are lacking. Are those CAFs ?

As stated by the authors, ECM production and remodeling is primarily performed by CAFs. This may suggest that the most important role of BNC2 in cancer cells could relate to the control of EMT while regulation of ECM-related activities would be relevant to BNC2 in CAFs. This issue needs to be discussed by the authors and better put into perspective when drawing conclusions.

Statistics:

t-tests are not appropriate for defining statistical significance of all data types presented
Number of replicates should be indicated

Referee Cross-Comments

I concur with reviewer 3's point 2. The paper needs clarification with regards to cells in which BNC2 operates.

Reviewer #2 (Comments to the Authors (Required)):

This is an interesting study by Orang et al highlighting the association between BNC2 expression and epithelial and mesenchymal properties of breast cancer cells. The authors present data showing that BNC2 is a prominent regulator of the extracellular matrix, including composition and degradation, and they correlate the expression of BNC2 with invasive and migratory properties of breast cancer cells. Regarding clinical significance, although BNC2 is associated with aggressive behavior in vitro, it is not clearly associated with overall survival or other clinical progression benchmarks.

The quality of the data presented in the manuscript is generally impressive (particularly, the validation of the results across multiple cell culture models, using gain and loss of function approaches). However, there seems to be multiple incomplete stories presented here, for instance: the role of BNC2 in driving expression of both mesenchymal and epithelial markers, the potential regulation of BNC2 by miR-200, the regulation of BNC2 of matrix remodeling, the role of BNC2 in both fibroblasts and cancer cells, the differential expression of BNC2 in different cancer and normal tissue, the clinical relevance of BNC2 in different cancer types. Despite the solid data, the work does not advance our understanding of how BNC2 is promoting (or suppressing) cancer progression. A more coherent model should be examined.

Specific comments:

Figure 1

In Figure 1A, it seems that most tumors expressing high BNC2 have both high mesenchymal and epithelial scores. Statistical analysis should be done to properly assess the correlation between the three variables, BNC2, mesenchymal score and epithelial score.

In Figure 1C, it would be more helpful to assess BNC2 in the different breast cancer subtypes, independently of the mesenchymal score (i.e., combining the high and low Mes and providing statistical analysis). Examining the data in the TCGA dataset (both in Gene Expression Profiling Interactive Analysis and in cbiportal), it seems that basal tumors have the lowest expression and luminal A the highest. This is an interesting point that should be discussed since it goes against the association of BNC2 expression with mesenchymal differentiation as luminal A tumors are more epithelial. On the other hand, the authors should consider additional datasets; a quick examination of the Metabric dataset (downloaded from cbiportal) showed that claudin-low tumors actually exhibit equally high levels of BNC2 similar to luminal A tumors. This is very interesting since expression of the miR-200 family has been shown by Rosen and collaborators (and validated by the Moorhead lab) to be extremely low in claudin-low breast cancer. Together with the data presented in Figure 2, these analyses suggest that BNC2 seems to be elevated in both very epithelial and very mesenchymal tumors. Staining of BNC2 in different subtypes, for instance in TNBC and HR positive, which represent two examples of basal and luminal tumors, respectively, would be beneficial for examining whether the protein is differentially expressed and more importantly to assess whether it is enriched in stromal cells, such as fibroblast, versus cancer cells.

Supplementary Figure 1

The Sigma antibody (cat# HPA018525) presented in Supplementary Figure 1 seems to be well validated in The Human Protein Atlas. It could be worth testing for IHC staining.

Regarding the antibody validation, could the authors assess the different antibodies in cell lines with high endogenous levels of BNC2 instead of T47D and MCF7 cells exogenously expressing it?

is there an explanation why T47D cells express FLAG-BNC2 to much lower levels compared to MCF7 cells?

Figure 2

It would be beneficial to examine the effects of over expressing BNC2 in a mesenchymal cell line with relatively low endogenous expression, such as MDA231 cells.

Figure 3

Are there any transcription co-regulators of BNC2 that were highlighted in the analysis? Ingenuity Pathway Analysis (IPA) would be revealing.

Figure 4

Very impressive data showing that BNC2 potentially regulates the expression of matrix remodeling genes.

Figure 5

Quantification of fluorescence intensity could be performed to assess the expression of the collagen I and V. Alternatively, validation by western blot could be performed. Scale bars are missing.

It would be beneficial to examine cells at lower confluence.

Figure 6

In Figure 6C, the organization of the collagen and the actin fibers seems more parallel in the BNC2 knockdown cultures. It would be beneficial to quantify that. It is unexpected that this matrix organization was associated with decreased cancer cell migration (in fact, parallel matrix organization has been shown by the Keely lab, and validated by others, to promote more migration). A possible explanation is that adhesion is much higher in the cells cultured on matrix derived from the knock down fibroblasts. The authors could examine this by analyzing focal adhesion and quantifying adhesion strength.

Are the multiple FITC panels in 6H just more examples of the same condition? Needs better labeling. Scale bars are not clearly labeled and the magnification seems different in the last two panels.

Figure 7

It is not clear what are the assays and conditions used in the presented experiments. Individual data points and example images of the cells should be presented.

Figure 8

It would be more beneficial to compare metastatic vs non-metastatic tumors instead of normal to cancer (also, a clarification of what is considered normal should be provided).

For the OS plots, the authors should also compare the top and bottom quartiles.

Supplementary Figure 7. There is no point in comparing OS across all cancers combined; it is unusual to group cancers from different tissue types together.

Supplementary Figure 8

The data in this figure are subpar since it only includes 80 months of observation (not sufficient for examining long term recurrence in breast cancer).

Reviewer #3 (Comments to the Authors (Required)):

This paper reports the function of basenuclin 2 in mesenchymal cells and the relation between levels of BNC2 and cancer invasiveness. The first part of the paper deals with the normal function of BNC2 in the regulation of the extracellular matrix. The authors report convincingly that BNC2, which is highly expressed in mesenchymal cells, regulates both the composition and the degradation of the extracellular matrix. In the second half of the paper the authors demonstrate that BNC2 expression correlates with tumor progression and poor prognosis. They attribute this to the fact that greater levels of BNC2 facilitate invasion by altering the ECM. There appear to be two drawbacks as to this conclusion :

1. BNC2 levels are lower than normal and not higher in a number of human cancers, particularly ovarian cancer, in which the first association between BNC2 and cancer was reported.
2. The cell type in which the tumor-promoting activity of BNC2 operates is not clear. Most human tumors are epithelial and BNC2 is mostly mesenchymal. Human solid tumors are often heterogeneous. Where is BNC2 operating : in the epithelial tumor cells ? in the epithelial cells undergoing mesenchymal transition ? in the supporting fibroblasts ? It would appear worthwhile to examine human tumors expressing high levels of BNC2 by immunocytochemistry in order to obtain a clearer picture of what is happening in the tumor at the cell level. It is difficult to understand how the same protein could be both a tumor-suppressor at early stages and a tumor-promoter at late stages. Some kind of plausible explanation should be given.

We would like to thank the reviewers for their thorough assessment and considered opinions. We will answer each point raised individually however before we do, we will provide a short summary of the major changes. Each of the figures that are referenced in this paragraph represent new data not in the original submission. Several reviewers noted there was uncertainty over the cell population in which BNC2 was expressed in cancer. We present evidence that it is in fibroblasts in which BNC2 is most highly and widely expressed (especially Figures 8D,E) and that the confusion regarding whether BNC2 plays a tumour promoting or suppressing role based on patient survival data is perhaps best understood as BNC2 serving as a marker of infiltrating fibroblasts more so than a direct marker of the cancer cells themselves. As such, BNC2 expression in Kaplan-Meier survival data parallels that of FAP, a well established marker of fibroblast activation which is increased in higher tumour grades and is especially abundant in bulk-seq samples from patients with poor prognosis (Figures 8C,F, Supp. Figs 10,11). We also provide further analysis to demonstrate that despite being mesenchymal in its own expression (Supp. Fig 2D), it is not itself a driver of epithelial/mesenchymal gene expression programs (Figure 2F). Further analysis is provided regarding the expression of BNC2 in breast tumours (Figures 1B,C), quantitation of collagen expression in response to BNC2 knockdown (Figure 5C) and examination of whether the migratory/invasive effects of BNC2 manipulation are likely attributable to changes in proliferation/cell number (Supp. Fig 9). Further characterization of commercial BNC2 antibodies (Supp. Fig 1), representative images from migration/invasion assays (Supp. Fig 7) and data from independent clonal cell lines (Supp. Fig 8) is also now included.

Reviewer #1 (Comments to the Authors (Required)):

Orang et al report that BNC2 controls ECM composition and degradation by breast cancer cells and CAFs. The authors further identified that BNC2 modulates cancer cell motility and invasion properties in line with BNC2 expression being linked to poor patient prognosis.

Many data were obtained using stable cell-lines. However, if data were made robust through the use of different clones is not stated.

Multiple clonal cell lines have been established and analysed during this work which is now made clear in the text and additional supplementary data.

In Fig 2D, data from 2 clones are included which show variable expression of one mesenchymal and one epithelial marker gene in response to BNC2 induction. This is included within a larger set of data to indicate that BNC2 manipulation does not promote a consistent epithelial or mesenchymal gene expression program (as one would expect of a factor that regulates EMT/MET).

In new Supplementary Figure 8, data from independent stable cell lines is included which shows reproducible effects between clones with regard to the effect of BNC2 knockdown or induction on cell motility. Supp. Fig 8A is matched to Figure 7H. Supp. Fig 8B is matched to Figure 7J.

Greater details regarding shBNC2 used are needed.

Our apologies. Sequence of the shRNAs are now included in Supplementary Table 3. The methodology for generating inducible shBNC2 cell lines is described in materials and methods.

Several siRNA for BNC2 are reported in table S3 but how they were used in the different experiments and whether consistent data were obtained is not described.

In the materials and methods section, we now specify the concentration and usage of siRNAs, achieving knockdown through the equimolar mixing of 3 Genepharma or 2 Qiagen-derived siRNAs (all independent sequences), then comparing results to an equivalent amount of control siRNA. The loss of BNC2 expression in 3

breast cancer cell lines corresponds with decreased migratory and invasive capacity. This has been shown using both Genepharma and Qiagen siRNAs independently (as specified in the Figure 7 legend), with results further supported by an independent shRNA, and by the opposing effect of BNC2 induction in both iBNC2 MCF7 and T47D stable cell lines (Figure 7). As we now make clear in the materials and methods section, Genepharma-derived siRNAs were used to prepare BNC2 knockdown samples for RNA-sequencing. qPCRs to confirm effects on collagens, matrix metalloproteases and other core EMT-associated genes were also performed using Genepharma siRNAs (Figures 2, 4 and 6).

The authors need to check that on results obtained in different assays such as wound healing and migration assays are not biased by an effect of BNC2 on cell proliferation.

We agree with this suggestion and have included a new supplementary figure (Supp. Fig 9) to address this comment directly. The siRNAs did decrease total cell number, but the size of the effect is small and does not provide an explanation for the large effects on motility that we show in Figure 7. The shRNA and iBNC2 cells showed no change to cell number, yet both also show clear effects on wound closure capacity.

The authors state that BNC2 is not a general promoter of EMT when analyzing individual genes (p12) but latter define that BNC2 is a core component of an EMT expression module when mining RNAseq data (p13). These data and conclusions should be reconciled. How does data from fig2 and 4 compare?

BNC2 is a mesenchymally-expressed gene, but not an enforcer or driver of the mesenchymal phenotype, at least as this is typically defined by the expression of key marker genes. The new evidence we present to show this are as follow:

Reassessment of the correlation of BNC2 with the epithelial and mesenchymal status of breast tumours (Figure 1A, new Fig 1B,C) and cancer cell lines (new Supp. Fig 2D) further demonstrates that the expression of BNC2 is strongly correlated with that of other mesenchymal signature genes. The lack of consistent pro-MET gene expression in RNA-seq after BNC2 knockdown (new Fig 2E) indicates BNC2 does not promote or maintain mesenchymal gene expression. This is true across a 146 gene epithelial/mesenchymal signature set, thus expanding our assessment beyond the qPCR of key standard EMT/MET marker genes (Fig 2A-D, which also did not show consistent trends in epithelial and mesenchymal gene expression in response to BNC2 perturbation).

Regulation of ECM genes including collagen seems at odds with previous reports (e.g. Ref 51) but this is not discussed. Details regarding the fibroblasts used in this study are lacking. Are those CAFs ?

There is now extensive discussion comparing the similarities and differences between our findings and those of Bobowski-Gerard in the discussion section (discussion paragraph #5). This includes speculating on why in some circumstances, BNC2 knockdown might have a positive effect on collagen expression, whilst in other contexts, the loss of BNC2 decreases the expression of collagen genes.

Full details of the mammary fibroblasts used in this study are now included in materials and methods. The fibroblasts used are not specifically CAFs, though we note that in tumours, BNC2 is primarily expressed in CAFs (new Fig. 8E). We have performed an experiment in which we treated our mammary fibroblasts with TGF β , resulting in a 4 fold upregulation of the activation marker SMA. BNC2, at least at the RNA level, remained unaffected by further TGF β exposure (unlike in breast cancer cells in Fig.1G-I). This suggests BNC2 may already be maximally expressed in the mammary fibroblasts. We have not included this data within the current manuscript.

As stated by the authors, ECM production and remodeling is primarily performed by CAFs. This may suggest that the most important role of BNC2 in cancer cells could relate to the control of EMT while regulation of ECM-

related activities would be relevant to BNC2 in CAFs. This issue needs to be discussed by the authors and better put into perspective when drawing conclusions .

We agree. Please see response to the “referee cross-comment” below and our response to referee 3.

Statistics:

t-tests are not appropriate for defining statistical significance of all data types presented. Number of replicates should be indicated.

The description of our statistical tests have been expanded in Materials and Methods. Unpaired, two-tailed t-tests were performed for all qPCR, luciferase and other cell based assays (migration, invasion etc) as we are making a series of individual comparisons between BNC2 induction/knockdown and cells transfected with control RNA. Correlation analyses associated with Fig 1A-C and Supp. Fig 2A-B are Pearson correlations as now described. Enrichment of gene ontologies are calculated within the associated bioinformatic packages.

Referee Cross-Comments

I concur with reviewer 3's point 2. The paper needs clarification with regards to cells in which BNC2 operates.

We are also in full agreement with this comment. To address, we have included new data, especially that presented in Figures 8D and 8E that demonstrate the cells in tumours that are expressing BNC2 are primarily fibroblasts (and not the cancer cells themselves). At the end of both the results and discussion section, we now discuss that BNC2, which is elevated at later cancer stages or in patients with poorer prognoses, is most likely a marker for infiltrating fibroblasts rather than being upregulated with the cancer cells themselves as we had initially assumed. High levels of fibroblast infiltration are known to be associated with poor prognosis. Please also see comments that address referee #3.

Reviewer #2 (Comments to the Authors (Required)):

This is an interesting study by Orang et al highlighting the association between BNC2 expression and epithelial and mesenchymal properties of breast cancer cells. The authors present data showing that BNC2 is a prominent regulator of the extracellular matrix, including composition and degradation, and they correlate the expression of BNC2 with invasive and migratory properties of breast cancer cells. Regarding clinical significance, although BNC2 is associated with aggressive behavior in vitro, it is not clearly associated with overall survival or other clinical progression benchmarks.

The quality of the data presented in the manuscript is generally impressive (particularly, the validation of the results across multiple cell culture models, using gain and loss of function approaches). However, there seems to be multiple incomplete stories presented here, for instance: the role of BNC2 in driving expression of both mesenchymal and epithelial markers, the potential regulation of BNC2 by miR-200, the regulation of BNC2 of matrix remodeling, the role of BNC2 in both fibroblasts and cancer cells, the differential expression of BNC2 in different cancer and normal tissue, the clinical relevance of BNC2 in different cancer types. Despite the solid data, the work does not advance our understanding of how BNC2 is promoting (or suppressing) cancer progression. A more coherent model should be examined.

We thank the reviewer for these comments. Based on this, and other reviewer comments, we have included significant new data and discussion which has clarified our thinking that BNC2 primarily serves as a regulator of the matrix in fibroblasts and it is the expression of BNC2 in fibroblasts that most likely explains the association of high BNC2 expression with high cancer grade and poor patient prognosis – high levels of fibroblast infiltration are well known to be associated with such conditions. Our main evidence to support

these claims is that BNC2 is primarily expressed in fibroblasts in cell lines (Fig. 8D), normal tissue (Fig. 6A) and tumours (Fig. 8E) and that the poor patient survival in the cohorts with the highest levels of BNC2 expression very closely resemble patients with the highest levels of FAP expression (Fig. 8C,F), a marker of fibroblast infiltration. This is true for breast cancer, and at least a number of other cancer types (Supp Fig. 10C,D). This is also broadly consistent with a recent report by Bobowski-Gerard (Nat Comm, 2022), that also implicates BNC2 as a matrisomal regulator in myofibroblasts that are associated with liver fibrosis.

BNC2 is particularly highly expressed in BT549 breast cancer cells (Fig. 8D) and we find it is capable of regulating a similar cohort of matrisome and secretome genes (Fig. 4,5, Supp Fig. 6), though for the reasons described above, we now believe its role is primarily in fibroblasts.

We demonstrate that BNC2 can be regulated by miR-200 (Fig. 1J-L) as one might expect of a mesenchymal-gene, however we show that BNC2 itself does not promote a mesenchymal phenotype as indicated by the expression of EMT genes that do not consistently change in response to BNC2 perturbation (Fig. 2A-F). As such, we make no claim that BNC2 is itself a regulator of mesenchymal gene expression and are instead clear to state it does not appear to be another EMT-regulating gene in our discussion of Figure 2.

Specific comments:

Figure 1

In Figure 1A, it seems that most tumors expressing high BNC2 have both high mesenchymal and epithelial scores. Statistical analysis should be done to properly assess the correlation between the three variables, BNC2, mesenchymal score and epithelial score.

In Figure 1C, it would be more helpful to assess BNC2 in the different breast cancer subtypes, independently of the mesenchymal score (i.e., combining the high and low Mes and providing statistical analysis). Examining the data in the TCGA dataset (both in Gene Expression Profiling Interactive Analysis and in cBioportal), it seems that basal tumors have the lowest expression and luminal A the highest. This is an interesting point that should be discussed since it goes against the association of BNC2 expression with mesenchymal differentiation as luminal A tumors are more epithelial. On the other hand, the authors should consider additional datasets; a quick examination of the Metabric dataset (downloaded from cBioportal) showed that claudin-low tumors actually exhibit equally high levels of BNC2 similar to luminal A tumors. This is very interesting since expression of the miR-200 family has been shown by Rosen and collaborators (and validated by the Moorhead lab) to be extremely low in claudin-low breast cancer. Together with the data presented in Figure 2, these analyses suggest that BNC2 seems to be elevated in both very epithelial and very mesenchymal tumors. Staining of BNC2 in different subtypes, for instance in TNBC and HR positive, which represent two examples of basal and luminal tumors, respectively, would be beneficial for examining whether the protein is differentially expressed and more importantly to assess whether it is enriched in stromal cells, such as fibroblast, versus cancer cells.

To address the question of epithelial vs mesenchymal expression, we performed a new analysis (new Fig 1B,C) that clearly indicates BNC2 is specifically expressed in tumours that have a strong mesenchymal signature. There is no relationship between BNC2 expression and that of epithelial genes. By plotting mesenchymal and epithelial signature gene sets separately, and including a statistical correlation test, this observation is easier to interpret. Further new evidence to support the co-expression of BNC2 with a mesenchymal gene signature is shown in new Supp Fig 2D where compared to a panel of 146 E/M marker genes, there is a strong positive association of BNC2 expression with mesenchymal genes and a strong negative association with epithelial genes.

New Fig 2F is included to demonstrate that BNC2 knockdown does not lead to the promotion or inhibition of either mesenchymal or epithelial genes en masse. This has been included to accompany Figure 2 A-D in which there is little coherence between the regulation of key mesenchymal or epithelial marker genes in response to BNC2 knockdown, as assessed by qPCR.

The suggestion to probe BNC2 expression more comprehensively with respect to breast cancer subtype is valid, however our efforts to address other reviewer comments have reshaped our conclusions as we now know the vast majority of BNC2 expression comes from the fibroblasts that are associated with the tumour, and not from the cancer cells themselves (new Fig.8D-E). The Kaplan-Meier plots for example are probably best interpreted as high BNC2 (like high FAP) acting as a marker for infiltrating fibroblasts, rather than reflecting differences between breast cancer cell subtypes (Fig. 8C,F; Supp Fig. 10C,D). We now describe this in detail in paragraph #5 of the discussion.

Supplementary Figure 1

The Sigma antibody (cat# HPA018525) presented in Supplementary Figure 1 seems to be well validated in The Human Protein Atlas. It could be worth testing for IHC staining.

Regarding the antibody validation, could the authors assess the different antibodies in cell lines with high endogenous levels of BNC2 instead of T47D and MCF7 cells exogenously expressing it?

Is there an explanation why T47D cells express FLAG-BNC2 to much lower levels compared to MCF7 cells?

We have extensively tested the available antibodies in IHC, now included as additional data in Supp Fig 1. In no instance was staining reduced by siRNA-mediated BNC2 knockdown, despite the siRNAs clearly being capable of reducing BNC2 mRNA levels as shown by qPCR. Western blotting of proteins derived from 2 cell lines that express abundant BNC2 mRNA (using the sigma antibody) shows a clear band at a size that is approximate to BNC2, but this band is not knocked down by BNC2 siRNAs. Our laboratory has extensive experience with both IHC and westerns and whilst we do note that the antibody has been published and used elsewhere, we are simply unable to demonstrate that BNC2 is the primary protein that is being picked up, at least in our hands. For this reason, we are not reporting endogenous BNC2 protein expression within our paper. We do not know why FLAG-BNC2 is expressed at higher levels in MCF7 cells compared to T47Ds, but the observation is consistent between protein and RNA level. This may be a clonal issue, or it may reflect a genuine biological difference between these cells.

Figure 2

It would be beneficial to examine the effects of over expressing BNC2 in a mesenchymal cell line with relatively low endogenous expression, such as MDA231 cells.

Given our growing realization of the importance of endogenous BNC2 in fibroblasts, we have not performed this experiment.

Figure 3

Are there any transcription co-regulators of BNC2 that were highlighted in the analysis? Ingenuity Pathway Analysis (IPA) would be revealing.

We are yet to fully explore the factors that regulate BNC2, or the co-factors with which BNC2 interacts to regulate the expression of other genes. We are commencing this work – looking for BNC2 interacting proteins and analysing differentially expressed genes among fibroblast sub-types that do or do not endogenously

express BNC2 (found in single cell sequencing; unpublished data obtained through collaborators). However, this is beyond the scope of this report.

Interestingly, of the 164 genes most closely correlated with BNC2 expression (genes with >0.7 connectivity in the “red” module – WGCNA analysis in Fig.3), 14 are transcription factors. These include PRRX1 and TBX5, noteworthy as Da Silveira et al, Sci Rep 2017, used transcriptomic data to propose a transcriptional module containing BNC2 and 4 other mesenchymal transcription factors, of which PRRX1 and TBX5 were a part. We hope our ongoing efforts to explore this area in greater detail will be a subject of future publication.

Figure 4

Very impressive data showing that BNC2 potentially regulates the expression of matrix remodeling genes.

Thank you for the comment.

Figure 5

Quantification of fluorescence intensity could be performed to assess the expression of the collagen I and V. Alternatively, validation by western blot could be performed. Scale bars are missing. It would be beneficial to examine cells at lower confluence.

Scale bars on immunofluorescence figures and IHC quantitation (Figure 5c) have now been included. Results are consistent across images regardless of local confluency.

Figure 6

In Figure 6C, the organization of the collagen and the actin fibers seems more parallel in the BNC2 knockdown cultures. It would be beneficial to quantify that. It is unexpected that this matrix organization was associated with decreased cancer cell migration (in fact, parallel matrix organization has been shown by the Keely lab, and validated by others, to promote more migration). A possible explanation is that adhesion is much higher in the cells cultured on matrix derived from the knock down fibroblasts. The authors could examine this by analyzing focal adhesion and quantifying adhesion strength.

Are the multiple FITC panels in 6H just more examples of the same condition? Needs better labeling. Scale bars are not clearly labeled and the magnification seems different in the last two panels.

We apologise that the images were not more clearly labelled. The second image was included to show a larger field of view at lower magnification. We have tried to streamline the new figure and have removed the image with the lower magnification. This is not necessary as the quantitation of gelatin degradation (Fig.6H) contains this information across a large number of cells (a minimum of 250 cells per variable as described in materials and methods). Figure 6D (collagen from a cell-derived matrix), shows increased collagen deposition after BNC2 knockdown, though to our eye at least there is no obvious effect on its organization. Differential adhesion to these matrices may explain our observation of altered cell motility, though we have not further examined this possibility.

Figure 7

It is not clear what are the assays and conditions used in the presented experiments. Individual data points and example images of the cells should be presented.

Example images of migration, invasion and wound healing assays (that are presented in Figure 7) are also now shown in Supplementary Figure 7. Individual data points are now included throughout the figures.

Figure 8

It would be more beneficial to compare metastatic vs non-metastatic tumors instead of normal to cancer (also, a clarification of what is considered normal should be provided).

For the OS plots, the authors should also compare the top and bottom quartiles.

The OS plots compare survival between BNC2 expression broken into high : low 50%/50%; 75%/25% and 90%/10%. More thorough characterization of what is considered “normal” is limited by the data and information from which GEPIA draws. OS plots with FAP is now also included to provide circumstantial support to the hypothesis for which we now argue – the association of BNC2 expression with patient survival is indicative of the presence of infiltrating fibroblasts (see Fig. 8E) rather than reflecting differences in the cancer cells themselves.

Supplementary Figure 7 (new Supplementary Fig. 10). There is no point in comparing OS across all cancers combined; it is unusual to group cancers from different tissue types together.

We agree. We have removed this analysis from the supplementary data.

Supplementary Figure 8

The data in this figure are subpar since it only includes 80 months of observation (not sufficient for examining long term recurrence in breast cancer).

We agree that the time of observation limits the conclusions that can be drawn. It is the data that was available using kmplot and is included as a supplementary figure to support other conclusions in the paper. We draw no additional conclusions solely using this data. We can remove if necessary, though we would rather keep as supplementary data as it provides further supporting evidence of higher BNC2 expression being associated with poorer outcome in conditions of increasing grade of breast cancer (again, likely due to fibroblast infiltration).

Reviewer #3 (Comments to the Authors (Required)):

This paper reports the function of basonudin 2 in mesenchymal cells and the relation between levels of BNC2 and cancer invasiveness. The first part of the paper deals with the normal function of BNC2 in the regulation of the extracellular matrix. The authors report convincingly that BNC2, which is highly expressed in mesenchymal cells, regulates both the composition and the degradation of the extracellular matrix. In the second half of the paper the authors demonstrate that BNC2 expression correlates with tumor progression and poor prognosis. They attribute this to the fact that greater levels of BNC2 facilitate invasion by altering the ECM. There appear to be two drawbacks as to this conclusion :

1. BNC2 levels are lower than normal and not higher in a number of human cancers, particularly ovarian cancer, in which the first association between BNC2 and cancer was reported.

2. The cell type in which the tumor-promoting activity of BNC2 operates is not clear. Most human tumor are epithelial and BNC2 is mostly mesenchymal. Human solid tumors are often heterogeneous. Where is BNC2 operating : in the epithelial tumor cells ? in the epithelial cells undergoing mesenchymal transition ? in the supporting fibroblasts ? It would appear worthwhile to examine human tumors expressing high levels of BNC2 by immunocytochemistry in order to obtain a clearer picture of what is happening in the tumor at the cell level. It is difficult to understand how the same protein could be both a tumor-suppressor at early stages and a tumor-promoter at late stages. Some kind of plausible explanation should be given.

Reviewer 1 also noted that a critical issue with the manuscript was the uncertainty regarding the cancer cell type in which BNC2 is expressed. We would like to have included immunocytochemistry to examine BNC2 expression within tumours, though as we present in an expanded Supp Fig 1, we do not believe the available antibodies are sufficiently specific. Nevertheless, this remains a critical question.

To examine further, we assessed new publically available data (new Fig. 8E) derived from the sequencing of >130K cells from 26 breast tumours (11 ER+, 5 HER2+ and 10 triple-negative; Wu et al, Nature Genetics 2022). This unequivocally indicated that the BNC2 that is associated with tumours, at least within this cohort, are specifically present in cancer-associated fibroblasts. Examination of cell lines (new Fig. 8D) further indicates that the expression of BNC2 within breast cancer cells is also likely to be a rare occurrence as very few breast cancer cell lines express BNC2 (BT549's used in this paper being a rare exception). As the reviewer points out, this is perhaps not surprising as cancers are generally epithelial whilst BNC2 is highly mesenchymal.

It is interesting that for most (though not all) cancer types, BNC2 is seemingly expressed at a lower level in the tumour (Fig. 8A, Supp Fig. 10A), though in those patients with particularly high BNC2, prognosis is worse (Fig. 8C, Supp Fig. 10C). BNC2 expression in tumours also increases with tumour stage (Fig.8B, Supp Fig. 10B). As we now describe in results and a re-written section of the discussion, we speculate the most likely explanation for these seemingly contradictory results is that the source of BNC2 being detected in bulk sequencing is not from the tumour cells themselves, but from associated fibroblasts and that it is a higher infiltration of these cells that is the most likely cause of poor prognosis (please refer to our first response to reviewer #2 detailing our rationale for a fibroblast-centric view of BNC2 fuction). In keeping with this, the Kaplan-Meier plots grouping patients by FAP expression (new Fig. 8F, Supp Fig. 10D), a fibroblast marker that is tightly co-expressed with BNC2, parallels results we find with BNC2. We speculate whatever influence BNC2 exerts on cancer most likely results from the regulation it exerts over the production and modulation of the matrisome by fibroblasts. This conclusion broadly agrees with that of Bobowski-Gerard et al, Nat Comm 2022 (we now specifically also discuss) who reported that BNC2 controls matrisomal gene expression in myofibroblasts that are associated with liver fibrosis.

July 10, 2023

RE: Life Science Alliance Manuscript #LSA-2023-01984-TR

Dr. Cameron Bracken
University of South Australia
Centre for Cancer Biology
Bradley Building, Frome Rd
Adelaide 5000
Australia

Dear Dr. Bracken,

Thank you for submitting your revised manuscript entitled "Basonuclin-2 regulates extracellular matrix production and degradation". We would be happy to publish your paper in Life Science Alliance pending final revisions necessary to meet our formatting guidelines.

- please address the final Reviewer 1 and 2's points
- please upload your Tables in editable .doc or excel format, and please be sure to label them accordingly
- please add ORCID ID for the corresponding (and secondary corresponding) author--you should have received instructions on how to do so
- please consult our manuscript preparation guidelines <https://www.life-science-alliance.org/manuscript-prep> and make sure your manuscript sections are in the correct order
- please move your main, supplementary figure, and table legends after the references section
- please ensure that all authors are entered in the Author Contribution section
- please add callouts for Figures 2F, 5C, S1A-C, S2D, S8A-B, S9A-C to your main manuscript text
- there is also a call out for Figure 9C, and there are only eight figures...please correct
- please add a conflict of interest statement to your main manuscript text

Figure checks:

- please indicate the scale bar size in Legend for Figure 6
- please revise the legends for Figures 2 and 6 so that the panels are introduced in order

A. FINAL FILES:

- An editable version of the final text (.DOC or .DOCX) is needed for copyediting (no PDFs).
- High-resolution figure, supplementary figure and video files uploaded as individual files: See our detailed guidelines for preparing your production-ready images, <https://www.life-science-alliance.org/authors>
- Summary blurb (enter in submission system): A short text summarizing in a single sentence the study (max. 200 characters)

including spaces). This text is used in conjunction with the titles of papers, hence should be informative and complementary to the title. It should describe the context and significance of the findings for a general readership; it should be written in the present tense and refer to the work in the third person. Author names should not be mentioned.

B. MANUSCRIPT ORGANIZATION AND FORMATTING:

Sincerely,

Reviewer #1 (Comments to the Authors (Required)):

The manuscript has been improved.

Regarding the main issue related to BNC2 expressing cells in tumors, the authors now stress out that BNC2 is primarily eThe manuscript has been improved.

Regarding the main issue related to BNC2 expressing cells in tumors, the authors now stress out that BNC2 is primarily expressed in CAFs and not the cancer cells themselves. This may make the initial part of the manuscript dedicated to the study of BNC2 in cancer cells awkward especially since the authors' main conclusion relate to extracellular matrix production and degradation.

If the authors were to split cancer cells based on their epithelial/mesenchymal phenotype in Fig.8D (similar to Fig.1F), would it show that BNC2 is high both in CAFs and in a subset of cancer cells with mesenchymal features ? This would then better justify the whole story.

Statistics not amended appropriately (e.g. Fig. 7J cannot appropriately be analyzed using t-tests).

Reviewer #2 (Comments to the Authors (Required)):

In this revised version of the manuscript the authors refocused the work on the role of BNC2 in the fibroblasts in the tumor microenvironment. They have addressed most of the comments raised in the initial review.

Minor comments:

The legend of Figure 5 is missing "C" (are the three data points representing different experiments?). Also in C, the label for the

Y-axis should read 488nm (not nM). there are other instances in the text where nm and μm should replace nM and μM . Regarding Supplementary Figure 8, there are other tools online that are available to generate similar plots with extended timeline (such as cbioportal). recurrence past the 6 year mark is high in ER positive breast cancer. Generating better plots would only make the data better.

Reviewer #3 (Comments to the Authors (Required)):

The authors have gone at great length to satisfy the reviewers. They have dealt satisfactorily with all my questions and I feel the paper is now ready for publication.

Dear Dr. Bracken,

Thank you for submitting your revised manuscript entitled "Basonuclin-2 regulates extracellular matrix production and degradation". We would be happy to publish your paper in Life Science Alliance pending final revisions necessary to meet our formatting guidelines.

-please address the final Reviewer 1 and 2's points

Each reviewer comment has been addressed point by point (see below).

-please upload your Tables in editable .doc or excel format, and please be sure to label them accordingly

Tables are now uploaded in excel format

-please add ORCID ID for the corresponding (and secondary corresponding) author--you should have received instructions on how to do so

ORCID IDs have been added. Please let us now if this has not been done correctly.

ORDID ID for Cameron Bracken = 0000-0001-7722-625X and for Katherine Pillman = 0000-0002-5869-889X.

-please consult our manuscript preparation guidelines <https://www.life-science-alliance.org/manuscript-prep> and make sure your manuscript sections are in the correct order

-please move your main, supplementary figure, and table legends after the references section

This has been done

-please ensure that all authors are entered in the Author Contribution section

This has been done. YKG not previously listed is now.

-please add callouts for Figures 2F, 5C, S1A-C, S2D, S8A-B, S9A-C to your main manuscript text

Specific reference to figures 2F and 5C are now added (apologies for the oversight). We have not added specific reference to any of the supplementary figure sub-parts. For example, Supp. Fig 1 is referenced as a whole rather than breaking it down into SF1a, SF1b etc. We have done this for all 12 supplementary figures and so we have not changed this in the resubmission to specifically cite those sections quoted. We can do so if important (please advise), but we're not sure it helps and the way we have cited supp data is at least consistent.

-there is also a call out for Figure 9C, and there are only eight figures...please correct

Apologies. This has been corrected.

-please add a conflict of interest statement to your main manuscript text

This has been done.

Figure checks:

-please indicate the scale bar size in Legend for Figure 6

The bar scale has now been added to the figure legend.

-please revise the legends for Figures 2 and 6 so that the panels are introduced in order

The legend for figures 2 and 6 were already presented in the correct order, though Figures 6F-H have been restructured to make this clearer.

We are not planning a press release.

To upload the final version of your manuscript, please log in to your account: <https://lsa.msubmit.net/cgi-bin/main.plex>

A. FINAL FILES:

B. MANUSCRIPT ORGANIZATION AND FORMATTING:

Sincerely,

Reviewer #1 (Comments to the Authors (Required)):

The manuscript has been improved.

Regarding the main issue related to BNC2 expressing cells in tumors, the authors now stress out that BNC2 is primarily expressed in CAFs and not the cancer cells themselves. This may make the initial part of the manuscript dedicated to the study of BNC2 in cancer cells awkward especially since the authors' main conclusion relate to extracellular matrix production and degradation. If the authors were to split cancer cells based on their epithelial/mesenchymal phenotype in Fig.8D (similar to Fig.1F), would it show that BNC2 is high both in CAFs and in a subset of cancer cells with mesenchymal features ? This would then better justify the whole story.

That's a good suggestion we had not considered – we have replaced figure 8D with an alternate version in which the breast cancer cell lines are now divided into Epithelial and Mesenchymal-like (there is a very clear demarcation between the two groups based on E-cadherin and ZEB1 expression). Most of the breast cancer cell lines within the CCLE are epithelial-like and almost all of these have levels of BNC2 which are either low or undetectable. Breast cancer cells that are more mesenchymal in their nature are much more likely to express high levels of BNC2 (as we would have predicted).

As we discuss on Page 15 of our manuscript (the results section where we transition from BNC2-expressing breast cancer cells to fibroblasts), transcriptomics indicates the ECM-modulating capacity of endogenous BNC2 is still operational in breast cancer cell lines in which it is expressed. This may or may not be consequential to the tumour microenvironment, but it did lead us to look in detail in fibroblasts where such a mechanism is vital. We then performed functional assays to show BNC2 is important here. Whilst not our original hypothesis, following the data led us to what we believe is an important insight regarding the role of BNC2 in fibroblasts which impacts tumour biology.

Statistics not amended appropriately (e.g. Fig. 7J cannot appropriately be analyzed using t-tests).

Figure 7J is a wound closure assay where we simply make direct comparison between no dox and + dox (-/+ BNC2 induction). This happens at 3 different time points. In each case, the comparison is between no dox and dox, and not between the time points themselves (which we now make clearer in the figure legend). As these are pairwise comparisons we believe t-tests are appropriate and require specific direction if this is deemed not to be the case. Having said that, the effect is clearly significant at 72hr, but relatively small, and very little hinges on this particular sub-figure.

Reviewer #2 (Comments to the Authors (Required)):

In this revised version of the manuscript the authors refocused the work on the role of BNC2 in the fibroblasts in the tumor microenvironment. They have addressed most of the comments raised in the initial review.

Minor comments:

The legend of Figure 5 is missing "C" (are the three data points representing different

experiments?). Also in C, the label for the Y-axis should read 488nm (not nM). there are other instances in the text where nm and μm should replace nM and μM .

Apologies. The legend to Figure 5 is now complete. nM here are replaced by nm. We have scanned the text and figures to ensure all similar instances have also been replaced. The axis label on the figure has also been corrected.

Regarding Supplementary Figure 8, there are other tools online that are available to generate similar plots with extended timeline (such as cbioportal). recurrence past the 6 year mark is high in ER positive breast cancer. Generating better plots would only make the data better.

We have re-evaluated Supplementary Figure 11 (old supp fig 8) and have retained the most important graphs derived from almost 3000 patients using KMplotter in which worse prognosis with high BNC2 expression is associated with higher tumour grade. We have also included additional datasets from Cbioportal, one of which shows a similar association derived from TCGA data, and another which does not find such an association. Additional data available in Cbioportal however showed that in both cases, there were differences in the classification of cancer types between low, moderate and high BNC2-expressing tumours, with high BNC2-expressing tumours more likely to be classified as metastatic or claudin-low, both of which are typically associated with poorer outcomes.

Reviewer #3 (Comments to the Authors (Required)):

The authors have gone at great length to satisfy the reviewers. They have dealt satisfactorily with all my questions and I feel the paper is now ready for publication.

July 20, 2023

RE: Life Science Alliance Manuscript #LSA-2023-01984-TRR

Dr. Cameron Bracken
University of South Australia
Centre for Cancer Biology
Bradley Building, Frome Rd
Adelaide 5000
Australia

Dear Dr. Bracken,

Thank you for submitting your Research Article entitled "Basonuclin-2 regulates extracellular matrix production and degradation". It is a pleasure to let you know that your manuscript is now accepted for publication in Life Science Alliance. Congratulations on this interesting work.

DISTRIBUTION OF MATERIALS:

Again, congratulations on a very nice paper. I hope you found the review process to be constructive and are pleased with how the manuscript was handled editorially. We look forward to future exciting submissions from your lab.

Sincerely,
